# Neuroendocrinology of the lung revealed by single-cell RNA sequencing

Christin S Kuo[1,2]*, Spyros Darmanis[3†], Alex Diaz de Arce[2], Yin Liu[2], Nicole Almanzar[1], Timothy Ting-Hsuan Wu[2], Stephen R Quake[3,4], Mark A Krasnow[2]*

[1]Department of Pediatrics, Stanford University School of Medicine, Stanford, United States; [2]Department of Biochemistry and Howard Hughes Medical Institute, Stanford University, Stanford, United States; [3]Department of Bioengineering, Stanford University, Stanford, United States; [4]Chan-Zuckerburg Biohub, San Francisco, United States

**Abstract** Pulmonary neuroendocrine cells (PNECs) are sensory epithelial cells that transmit airway status to the brain via sensory neurons and locally via calcitonin gene-related peptide (CGRP) and γ- aminobutyric acid (GABA). Several other neuropeptides and neurotransmitters have been detected in various species, but the number, targets, functions, and conservation of PNEC signals are largely unknown. We used scRNAseq to profile hundreds of the rare mouse and human PNECs. This revealed over 40 PNEC neuropeptide and peptide hormone genes, most cells expressing unique combinations of 5–18 genes. Peptides are packaged in separate vesicles, their release presumably regulated by the distinct, multimodal combinations of sensors we show are expressed by each PNEC. Expression of the peptide receptors predicts an array of local cell targets, and we show the new PNEC signal angiotensin directly activates one subtype of innervating sensory neuron. Many signals lack lung targets so may have endocrine activity like those of PNEC-derived carcinoid tumors. PNECs are an extraordinarily rich and diverse signaling hub rivaling the enteroendocrine system.

*For correspondence:
ckuo@stanford.edu (CSK);
krasnow@stanford.edu (MAK)

Present address: †Department of Microchemistry, Proteomics & Lipidomics, Genentech, California, United States

Competing interest: The authors declare that no competing interests exist.

## Editor's evaluation

The authors present a comprehensive profile of signals and sensors expressed in mouse and human PNECs by single-cell RNA sequencing. Analyses revealed a myriad combination of neuropeptide, neurotransmitter, receptor and channel genes in PNECs. The authors also surveyed cognate receptors expressed in epithelial cells, endothelial cells, stromal cells, immune cells, and pulmonary sensory neurons, identifying potential local targets for the PNECs signals. They showed one new signal, angiotensin II, directly activates a subpopulation of innervating pulmonary sensory neurons that project to the brain. The scRNA-seq profile of a lung carcinoid tumor suggests that selected PNECs are susceptible to carcinoid transformation and together, these data indicate that PNECs serve as sentinels to perceive multiple airway stimuli and express a variety of signals that either act locally, through pulmonary sensory neurons, or potentially through circulation to regulate homeostasis.

## Introduction

Pulmonary neuroendocrine cells (PNECs) are neuroepithelial cells scattered throughout the epithelium lining the airways, many as solitary cells and others in clusters of ~30 cells (called neuroepithelial bodies or NEBs) at airway branchpoints (*Scheuermann, 1997*). PNECs are thought to monitor airway

oxygen and respiratory status, and rapidly signal this internal sensory information via afferent sensory neurons to the brain to regulate breathing (*Garg et al., 2019*; *Nonomura et al., 2017*). Such signaling may be mediated by the classical neurotransmitter serotonin (*Lauweryns et al., 1973*), but PNECs also produce γ-aminobutyric acid (GABA) (*Schnorbusch et al., 2013*) and several neuropeptides including calcitonin gene-related peptide (CGRP), which can signal locally to lung goblet (via GABA, γ-aminobutyric acid) and immune cells (via CGRP, calcitonin gene-related peptide) and may contribute to asthma (*Barrios et al., 2019*; *Sui et al., 2018*). PNECs also function as reserve stem cells that repair the surrounding epithelium after injury (*Ouadah et al., 2019*; *Song et al., 2012*; *Stevens et al., 1997*), and they can be transformed by loss of tumor suppressor genes into the deadliest human lung cancer, small cell lung cancer (SCLC) (*Ouadah et al., 2019*; *Park et al., 2011*; *Song et al., 2012*; *Sutherland et al., 2011*), and likely other neuroendocrine tumors such as lung carcinoids accompanied by systemic symptoms from signals secreted by the tumor (*Davila et al., 1993*).

Despite the physiological and clinical significance of PNECs, molecular interrogation and understanding of PNEC function and diversity (*Mou et al., 2021*) has lagged because the cells are so rare, comprising just ~0.01% of human lung cells (*Boers et al., 1996*; *Travaglini et al., 2020*). Here we describe the isolation, expression profiling by single-cell RNA sequencing (scRNA-seq), and analysis of hundreds of mouse and human PNECs. The results reveal an extraordinary molecular diversity of these cells, including the expressed sensors along with dozens of neuropeptides and peptide hormones whose predicted targets indicate they transmit airway sensory information throughout the lung, to the brain, and potentially to the rest of the body.

## Results

### Enrichment and single-cell RNA sequencing of mouse pulmonary neuroendocrine cells

Because PNECs are among the rarest of lung cell types, they are not found, or only poorly represented, in lung scRNA-seq studies (*Han et al., 2018*; *Tabula Muris Consortium et al., 2018*; *Travaglini et al., 2020*, https://www.lungmap.net/). We therefore genetically labeled PNECs using a Cre recombinase-dependent fluorescent reporter (ZsGreen) by tamoxifen induction of *Ascl1*$^{CreER/+}$; *Rosa26*$^{LSL-ZsGreen/+}$ mice, and then depleted other, abundant lung cell populations and enriched for the labeled PNECs prior to scRNA-seq (*Figure 1A*, *Figure 1—figure supplement 1A and B*). Poly-adenylated mRNA from each sorted ZsGreen + or control cell was reverse transcribed and PCR-amplified using Smart-seq2 protocol (*Picelli et al., 2014*), and the obtained cDNA was used to generate libraries sequenced to a depth of $10^5$–$10^6$ reads/cell and quantified to determine expression levels of each gene in each cell. Cells with similar expression profiles were computationally clustered (*Butler et al., 2018*), and the cell type identity of each cluster was assigned based on expression of canonical lung cell type markers including *Ascl1*, *Calca*, and *Chga* for PNECs (*Figure 1—figure supplement 1C*). After filtering low quality cells and cell doublets, we obtained high quality transcriptomes (2025±450 genes (mean ± SD) detected per cell) of 176 PNECs. We also obtained 358 other pulmonary cells (*Figure 1—figure supplement 1D*) including *Ascl1* lineage-labeled glial cells.

We identified dozens of mouse PNEC markers (*Figure 1B and C*, *Supplementary file 1*) beyond the canonical four (*Calca* (encoding calcitonin gene-related peptide CGRP), *Ascl1*, *Syp* (synaptophysin), and *Chga* (chromogranin A)) (*Supplementary file 2*) using Wilcoxon rank sum (Seurat v2.3.4) to find differentially expressed genes selectively expressed by PNECs, then prioritizing those with high sensitivity (expressed in >85% of PNECs), specificity (<5% of the other lung cell types), and fold-difference in mean expression over other lung cell types. Many of the identified markers were expressed at higher levels (*Resp18*, *Pcsk1*, *Scg5*) and were more sensitive (*Resp18*, *Pcsk1*) and specific (*Chgb*, *Sez6l2*) than the canonical PNEC markers (*Figure 1C*). Among the top reported tracheal neuroendocrine cell (TNEC) markers, just over half (54% (19/35) *Montoro et al., 2018*; 63% (31/49) *Plasschaert et al., 2018* were also top PNEC markers, defining a set of shared airway neuroendocrine markers (*Supplementary file 1*). But there were also notable differences between TNECs and PNECs (e.g.) TNEC-selective *Cib3*, *Snca*, *Cxcl13*, *Mthfd2*) and PNEC-selective (*Ptprn*, *Piezo2*, *Igf2*, *Meg3*) (*Figure 1—figure supplement 1F*, *Supplementary file 1*), consistent with their different development, distribution, innervation, and functions (*Kuo and Krasnow, 2015*; *Montoro et al., 2018*; *Mou et al., 2021*; *Ouadah et al., 2019*; *Song et al., 2012*). Selective expression of the most robust PNEC

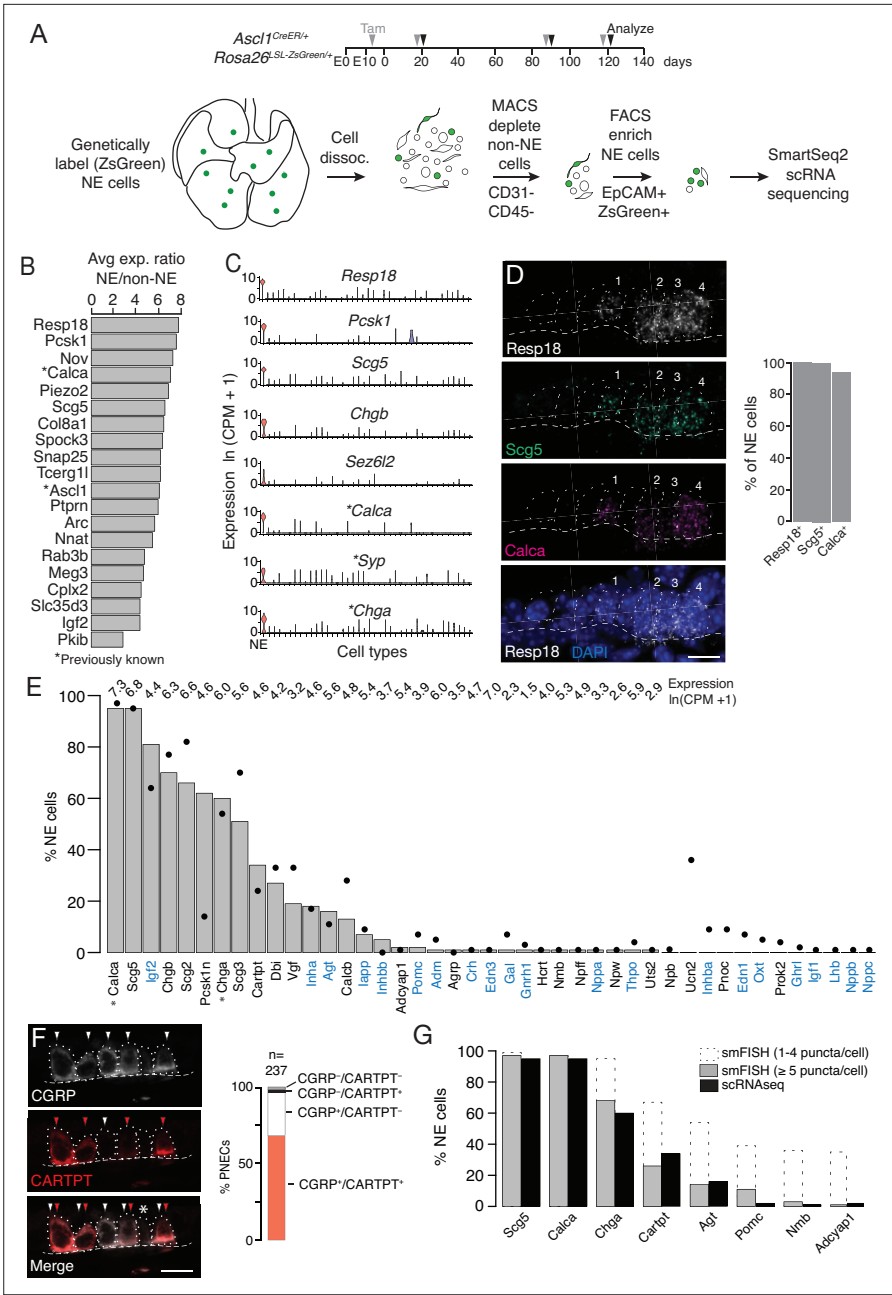

**Figure 1.** Single cell RNA sequencing of mouse PNECs reveals expression of dozens of peptidergic genes. (**A**) Strategy for labeling, enrichment, and scRNA-seq of exceedingly rare PNECs. Timeline (top) of tamoxifen (Tam) injections (gray arrowheads) of *Ascl1^CreER/+^;Rosa26^LSL-ZsGreen/+^* mice beginning at embryonic day (**E**) 13 - E14 to permanently induce ZsGreen expression in pulmonary neuroendocrine cells (PNECs or NE cells). Lungs were dissected at indicated ages (black arrowheads, postnatal day (PN) 21, PN90, and PN120) and mechanically and enzymatically dissociated (dissoc.) into single cells. Endothelial (CD31+) and immune (CD45+) cells were depleted by magnetic-cell sorting (MACS) then PNECs enriched by fluorescence-activated cell sorting (FACS) EpCAM⁺/Zsgreen⁺ double-positive cells. Sorted cells were analyzed by plate-based scRNA-seq using SmartSeq2 protocol. (**B**) Most sensitive and specific PNEC markers identified by scRNA-seq, ranked by ratio of the natural logs of the average expression (ln (counts per million, CPM + 1)) of the marker in PNECs (NE cells) vs. non-PNEC (non-NE) lung epithelial cells in mouse lung cell atlas (***Travaglini et al., 2020***). *, previously known PNEC marker. (**C**) Violin plots showing expression of five new markers (*Resp18, Pcsk1, Scg5, Chgb, Sez6l2*) and three previously known markers (*; *Calca, Syp, Chga*) across 40 cell types from mouse lung cell atlas (***Travaglini et al., 2020***). From left to right (x-axis): (1) neuroendocrine (NE, PNEC), (2) club, (3) multiciliated, (4) basal, (5) goblet, (6) alveolar type 1, (7) alveolar type 2, (8) glial, (9) smooth muscle, (10) myofibroblast, (11) adventitial fibroblast, (12) alveolar fibroblast,

*Figure 1 continued on next page*

*Figure 1 continued*

(13) pericyte, (14) mesothelial, (15) chondrocyte, (16) artery, (17) vein, (18) capillary aerocyte, (19) capillary-general, (20) lymphatic, (21) B cells, (22) Zbtb32+ Bcells, (23) plasma, (24) CD8+ T, (25) CD4+ T, (26) regulatory T, (27) Ly6g5bt + T, (28) natural killer, (29) Alox5+ lymphocyte, (30) neutrophil, (31) basophil, (32) alveolar macrophage, (33) interstitial macrophage, (34) plasmacytoid dendritic, (35) myeloid dendritic type 1, (36) myeloid dendritic type 2, (37) Ccr7+ dendritic, (38) classic monocyte, (39) nonclassical monocyte, (40) intermediate monocyte. (**D**) Close-up of neuroepithelial body (NEB) in PN155 wild type (C57BL/6NJ) mouse lung probed by multiplex single molecule RNA fluorescence in situ hybridization (smFISH) to detect expression of indicated PNEC markers, with DAPI nuclear counterstain. Dashed circles, individual PNECs (numbered); dashed line (basement membrane). Scale bar, 10 µm. Quantification (right) of clustered PNECs that express indicated markers (n=76 cells scored in left lobe and right lower lobe). Note classic marker *Calca* (CGRP) was not detected in 6% of *Resp18*⁺*Scg5*⁺double-positive PNECs. (**E**) Quantification of peptidergic gene expression in PNECs by scRNA-seq. Bars show percent of profiled PNECs (NE cells, n=176) with detected expression of the 43 peptidergic genes indicated; values above bars are log-transformed mean gene expression (ln (CPM + 1)) among expressing cells. Black dots, expression values from a second PNEC dataset (filled circles, n=92 PNECs) in which PNECs were genetically labeled using *Cgrp^{CreER}*;*Rosa26^{LSL-ZsGreen}* mice, sorted, and isolated on a microfluidic platform (***Ouadah et al., 2019***). (Comparison by Fisher's exact test (two-tailed) of the proportions of PNECs detected in the two scRNA-seq datasets is provided in ***Supplementary file 4***, with caveat that the comparison is of results from different techniques on different samples.) *, Previously known mouse PNEC peptidergic genes; blue highlight, classic hormone genes. (**F**) Micrograph of NEB from PN90 *Ascl1^{CreER/+}*;*Rosa26^{LSL-ZsGreen/+}* mouse lung immunostained for CGRP and newly identified PNEC neuropeptide CARTPT. White arrowheads, CGRP⁺ PNECs; red arrowheads, CARTPT⁺ PNECs; *, CGRP⁻ CARTPT double-negative PNEC. Right panel, quantification of CGRP and CARTPT staining in PNECs defined by *Ascl1*-CreER-lineage label (n=237 PNECs scored in three PN60 *Ascl1^{CreER/+}*;*Rosa26^{LSL-ZsGreen/+}* mice). (**G**) Quantification of PNEC (NE cell) expression of the indicated peptidergic genes by scRNA-seq (black bars, n=176 cells) and multiplex smFISH (grey bars and dashed extensions, n=100cells scored in NEBs from 2 mice, see ***Figure 2***). Grey bars, cells with high expression (≥5 puncta/cell); dashed extensions, cells with low expression (1–4 puncta/cell). Fisher's exact test (two-tailed) gave p=1 (not significant) for all comparisons of proportions of expressing PNECs for each gene as detected by smFISH (≥5 puncta/cell) vs. scRNA-seq (black bars); when the comparisons included cells with 1–4 puncta/cell by smFISH, differences were significant (p<0.05) for *Chga*, *Cartpt*, *Agt*, *Pomc*, *Nmb*, and *Adcyap1* but not for *Scg5* and *Calca* (p=0.9 for both).

The online version of this article includes the following source data and figure supplement(s) for figure 1:

**Source data 1.** Corresponds to *Figure 1G*.

**Figure supplement 1.** FACS enrichment strategy for PNECs.

markers was validated by single molecule in situ hybridization (smFISH) (*Resp18*, *Scg5*, *Figure 1D*) and immunohistochemistry (*Pcsk1*, see below).

## PNECs express dozens of neuropeptide and peptide hormone genes

PNECs were discovered by their hallmark secretory vesicles (***Scheuermann, 1997***) and later shown to secrete serotonin (***Lauweryns et al., 1972***), GABA (***Barrios et al., 2017***), and several neuropeptides including CGRP (***Johnson et al., 1988***; ***Scheuermann et al., 1987***) and chromogranin A (***Lauweryns et al., 1987***; ***Supplementary file 2***). To determine the full set of PNEC neurotransmitters and neuropeptides, we searched our scRNA-seq dataset for neurotransmitter biogenesis genes (***Supplementary file 3***) and neuropeptide and peptide hormone genes ("peptidergic" genes, ***Supplementary file 4***) expressed in PNECs. This confirmed that PNECs are GABAergic because many (28%) expressed biosynthetic gene *Gad1*. Although only rare (1%) PNECs express the serotonin biosynthetic gene *Tph*1 and none expressed *Tph2*, many (24%) expressed the reuptake transporter *Slc6a4*, suggesting they can modulate serotonin signaling through serotonin imported from other sources such as circulating serotonin (***Herr et al., 2017***). Some PNECs may also use other neurotransmitters because rare cells expressed key dopaminergic (*Th*, *Ddc*), glutamatergic (*Slc17a6*), cholinergic (*Slc18a3/*VAChT), or histaminergic (*Hdc*) genes (***Supplementary file 3***).

PNECs expressed dozens of genes that encode neuropeptides and/or peptide hormones. Thirty-one peptidergic genes were expressed in our dataset, and 30 of those plus an additional 12 were found in our second, independent dataset (see below), totaling 43 (47%) of the 91 peptidergic genes in mice (*Figure 1E*, *Supplementary file 4*). The full set includes previously known *Calca* (encodes CGRP) and *Chga* but also some of the most biologically and medically significant neuropeptide and peptide hormone genes (e.g. *Pomc* (pro-opiomelanocortin), *Hcrt* (hypocretin/orexin), *Agt* (angiotensinogen),

*Gnrh1* (gonadotropin releasing hormone), *Oxt* (oxytocin), *Lhb* (luteinizing hormone), *Crh* (corticotropin-releasing hormone), *Adm* (adrenomedullin), *Ghrl* (ghrelin), *Igf2* (insulin-like growth factor), *Inha*, *Inhba* and *Inhbb* (inhibins), *Edn1* and *Edn3* (endothelins), *Nppa* (atrial natriuretic peptide)). The seven granin genes (*Chga, Chgb, Scg2, Scg3, Scg5, Pcsk1n, Vgf*) and 16 others (*Calca, Calcb, Cartpt, Agt, Iapp, Adcyap1, Igf1, Igf2, Lhb, Inha, Inhba, Inhbb, Adm, Oxt, Pomc, Crh*) have also been detected in neuro-endocrine (NE) cells outside the lung, but this is the first description of NE cell expression for 17 PNEC peptidergic genes (*Dbi, Nppa, Nppb, Nppc, Nmb, Npb, Npff, Npw, Pnoc, Prok2, Gal, Gnrh1, Hcrt, Ucn2, Agrp, Thpo, Uts2*).

Some of the peptidergic genes including *Calca, Igf2,* and granins (*Chga, Chgb, Scg2, Scg3, Scg5, Pcsk1n*) were expressed in most PNECs analyzed, but others were detected in minor subpopulations (*Figure 1E*, *Supplementary file 4*). We analyzed peptidergic gene expression in our second scRNA-seq dataset of 92 PNECs that used a different strategy for PNEC labeling and capture (*Ouadah et al., 2019*) and obtained a similar distribution of peptidergic gene expression, with the exception of *Pcsk1n* (expressed in 14% of PNECs vs. 62% in original dataset) and *Ucn2* (34% vs 0%) (*Figure 1E*, *Supplementary file 4*). We also validated expression of eight peptidergic genes and determined the distribution and abundance of the expressing cells in vivo by immunostaining and multiplex smFISH. The canonical mouse PNEC marker and neuropeptide CGRP (*Calca*) was detected in 95% of PNECs by scRNA-seq, 95% of PNECs by immunostaining (n=237 scored cells in 3 mice) (*Figure 1F*), and 94% by smFISH (n=100 scored cells in 2 mice) (*Figure 1D*). *Cartpt* was detected in 34% of PNECs by scRNA-seq and twice that by immunostaining (68%) (*Figure 1F*) and smFISH (67%) (*Figure 1G*). smFISH also confirmed PNEC expression of the six other peptidergic genes examined (*Scg5*, *Chga*, *Agt, Pomc, Nmb, Adcyap1*) but generally gave higher percentages of expressed PNECs especially for the less abundantly expressed genes, implying smFISH is more sensitive than scRNA-seq in detecting gene expression (*Figures 1G and 2B*). Indeed, when PNECs with very low expression (1–4 mRNA puncta detected per cell) were not included in the smFISH quantification, there was excellent agreement between smFISH and scRNA-seq values (*Figure 1G*). The broadly expressed peptidergic genes *Calca, Scg5, Chga,* and *Cartpt* were detected in both solitary and clustered PNECs, and the four genes detected in smaller subpopulations (*Agt, Pomc, Nmb, Adcyap1*) were most commonly detected in NEBs at bronchial branchpoints, with local clustering observed within a NEB of the *Pomc*-expressing cells (*Figure 2—figure supplement 1A*).

We conclude that PNECs express dozens of peptidergic genes, nearly half (47%) of all annotated peptidergic genes and over an order of magnitude more than previously known, with most expressed in rare PNEC subpopulations. Statistical modeling of our PNEC sampling in scRNA-seq indicated we likely achieved saturation of PNEC peptidergic genes (*Figure 2—figure supplement 1B*), however the value of 43 expressed genes is a lower limit because even more sensitive methods of detecting gene expression in individual cells such as smFISH could identify additional PNEC peptidergic genes.

## PNECs express myriad combinations of peptidergic genes

The scRNA-seq analysis showed that individual mouse PNECs expressed 7.2 ± 1.9 (mean ± S.D.) peptidergic genes, with some cells expressing up to 13 (range 2–13). Remarkably, almost every PNEC expressed a different combination of peptidergic genes: 154 peptidergic patterns were identified among the 176 PNECs analyzed (*Figure 2A*). Because the number of combinations detected by scRNA-seq could be artificially inflated by 'technical dropout' (failure to detect an expressed gene), we also scored peptidergic combinations by the more sensitive technique of multiplex smFISH (*Figure 2B*). For the eight peptidergic genes probed in 100 PNECs, we identified 30 different cellular patterns of expression, similar to the 24 patterns detected by scRNA-seq for the same eight genes (*Figure 2C and D*). Thus, each PNEC expresses multiple peptidergic genes and in an extraordinary number of combinations.

The number and diversity of neuropeptides and peptide hormones expressed by PNECs is further expanded by post-transcriptional processing. Some of the expressed genes are alternatively spliced to produce transcripts encoding different neuropeptides or hormones with distinct expression patterns and physiological functions. For example, *Calca,* which encodes the classical PNEC neuropeptide CGRP, can be alternatively spliced to generate transcripts encoding the thyroid hormone calcitonin that regulates calcium homeostasis (*Amara et al., 1980*; *Amara et al., 1982*). Mapping of PNEC scRNA-seq reads at the *Calca* genomic locus revealed that *Calca* transcripts are alternatively spliced

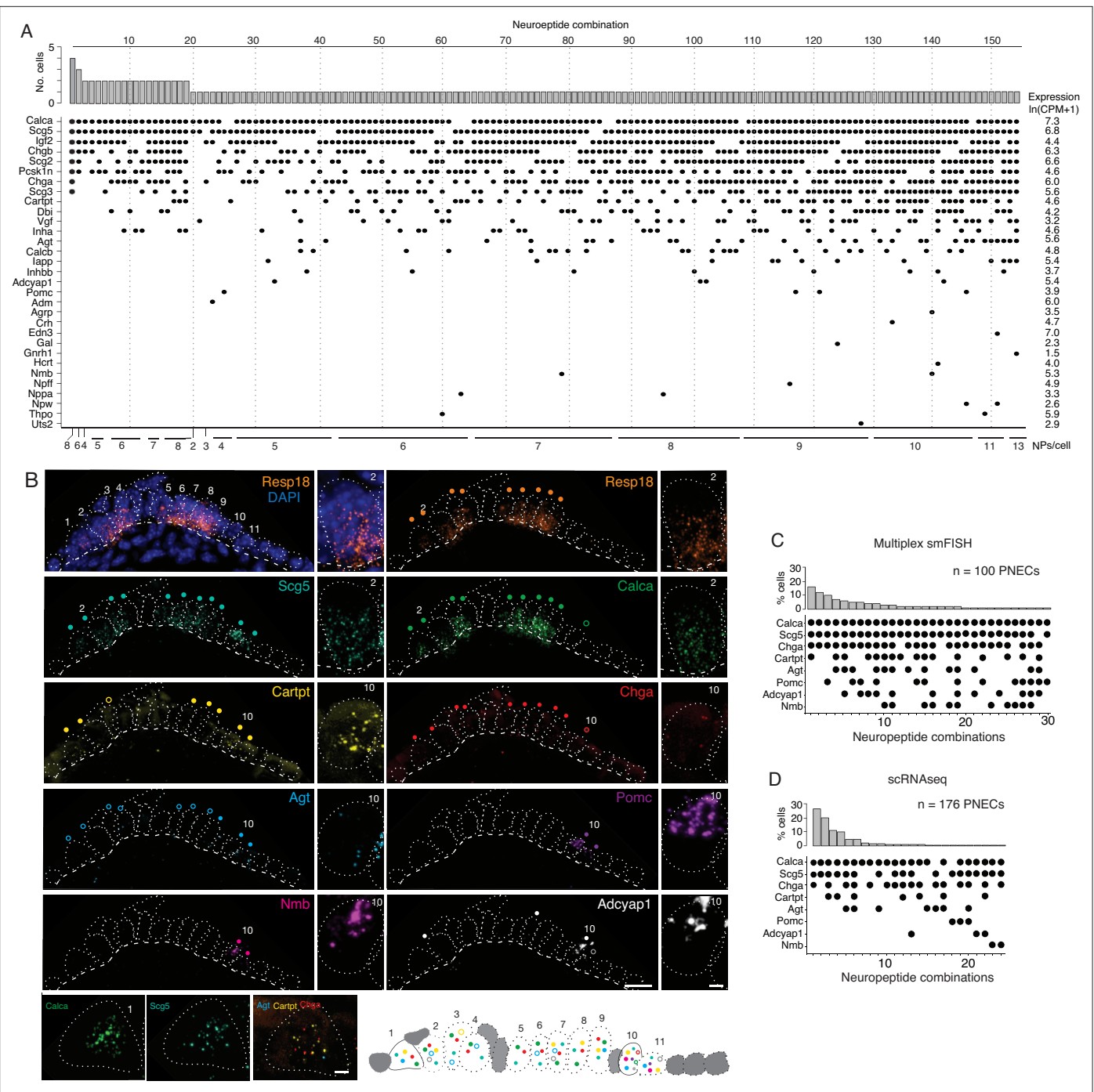

**Figure 2.** PNECs express myriad combinations of peptidergic genes. (**A**) Peptidergic genes expressed in individual PNECs (n=176) from scRNA-seq. Histogram (top) shows number of PNECs expressing each of the 154 observed combinations of expressed peptidergic (NP) genes (dots, bottom). Values at right are average expression level of peptidergic genes among expressing cells; values at bottom are number of expressed peptidergic genes for each combination (NPs/cell). Each PNEC expressed 7.2±1.9 (mean ± SD) peptidergic genes (range 2–13, median 7, mode 6). (**B**) Micrograph of NEB from adult PN155 wild type mouse lung probed by multiplex smFISH (RNAscope) for PNEC marker *Resp18* and the indicated peptidergic genes including ones detected by scRNA-seq in most (*Scg5, Calca*), some (*Cartpt, Chga*), or rare (*Agt, Pomc, Nmb, Adcyap1*) PNECs. Epithelial cells are outlined (white dots), basement membrane is indicated by dashed line, and individual PNECs numbered (n=11) with close-up of the indicated PNEC shown at right and PNEC 1 at bottom. Scale bar, 2μm. Filled colored circles above cells, PNECs with high expression of gene (≥5 puncta; filled circles); open circles, PNECs with low expression of gene (1–4 puncta, open circles). Schematic (bottom) shows summary of expression of the eight peptidergic genes (using colored circles as above) in the 11 PNECS in this NEB optical plane. Grey cells, other (non-PNEC) epithelial cells in field of view. Bar, 10μm. (**C**) Percent of PNECs expressing each of the 30 observed combinations of the eight peptidergic genes probed by smFISH (n=100 PNECs scored in

*Figure 2 continued on next page*

*Figure 2 continued*

NEBs from two wild-type mice). Filled circles, expressed peptidergic gene. (**D**) Percent of PNECs expressing each of the 24 observed combinations of the same eight peptidergic genes in scRNA-seq dataset (n=176 PNECs).

The online version of this article includes the following figure supplement(s) for figure 2:

**Figure supplement 1.** Peptidergic gene expression in PNECs.

in PNECs (*Figure 3A*); most cells expressed both calcitonin and CGRP transcripts, although the ratio varied greatly across individual PNECs and some cells exclusively expressed CGRP (20% of cells) or calcitonin (10%) (*Figure 3B*, *Figure 3—figure supplement 1A*). Similar results obtained by co-staining for CGRP and calcitonin proteins (*Figure 3C*, *Figure 3—figure supplement 1B*).

Many neuropeptide and peptide hormone mRNAs are translated as larger pre-pro-peptides that are proteolytically processed and modified to generate up to eight different neuropeptides and/or peptide hormones, typically with different expression patterns and functions (*Supplementary file 5*), which would further increase the PNEC signaling repertoire. A classic example is POMC, cleaved by proprotein convertases PCSK1 and PCSK2 to generate ACTH (adrenocorticotrophic hormone), MSH (melanocyte stimulating hormone), β-endorphin as well as others by additional processing events (*Figure 3D*). We surveyed expression of the processing enzyme genes in our scRNA-seq dataset (*Figure 3E*) and found that nearly all PNECs expressed *Pcsk1* (*Figures 1C and 3F*) and a subset of those expressed *Pcsk2*, predicting that POMC is processed in PNECs into multiple neuropeptides and peptide hormones including ACTH, α3-MSH, and β-endorphin. Confocal imaging of PNEC immunostains for POMC, CGRP, and calcitonin showed that each co-expressed neuropeptide or peptide hormone localized largely to its own secretory vesicles (*Figure 3G*), even ones expressed from the same gene (CGRP and calcitonin, *Figure 3C*), implying distinct vesicular targeting and packaging pathways. Thus, post-transcriptional and post-translational processing expands the number and diversity of peptidergic signals expressed by each PNEC, and their separate vesicular packaging raises the possibility of differentially regulated release and impact on targets.

## Diverse targets of PNEC signals

To predict the direct targets of the PNEC signals, we searched our molecular cell atlas of the lung, comprising the full expression profiles of nearly all lung cell types (*Travaglini et al., 2020*) and pulmonary sensory neurons (*Liu et al., 2021*), for cells that express the cognate receptors (*Supplementary file 5*). The only previously defined targets of PNEC signals are a subpopulation of immune cells (IL5 lineage-positive) proposed to be attracted to PNECs by secreted CGRP in a mouse model of asthma (*Sui et al., 2018*), and goblet cells, which are increased in macaque and mouse models of inflammation through neurotransmitter GABA from PNECs (*Barrios et al., 2019*; *Sui et al., 2018*). Recently, CGRP from tracheal TNECs has been found to support tracheal epithelial cells following hypoxic injury (*Shivaraju et al., 2021*).

Receptors for serotonin and GABA, the two major PNEC neurotransmitters, are expressed by the two pulmonary sensory neuron (PSN) subtypes that innervate NEBs (PSN4 (*Olfr78+*) and PSN7 (*Calb1+*); *Figure 4A*), identifying the first signals from PNECs to these afferent fibers that communicate pulmonary sensory information to the brain. GABA receptor genes were also expressed in goblet cells, supporting the conclusion from macaque and mouse models and identifying the specific GABA receptor subunit as GABRP, as well as in club cells, epithelial cells that neighbor PNECs (*Figure 4—figure supplement 1*). Receptors for the neurotransmitters glutamate, dopamine and histamine predicted by our transcriptomic data to be produced by rare PNECs (see above) were also expressed in innervating PSNs (*Figure 4A*) as well as in plasmacytoid dendritic cells (glutamate receptor *Grm8*) and basophils (histamine receptor *Hrh4*; *Figure 4—figure supplement 1B*).

Of the more than 90 neuropeptides and peptide hormones encoded by the 43 PNEC peptidergic genes, 36 have known receptors (*Supplementary file 5*). The lung expression patterns of these receptors are shown in *Figure 4B*, identifying dozens of lung cell types that can directly receive PNEC peptidergic signals. Indeed, every lung cell type expressed a receptor for at least one PNEC peptidergic signal, and most expressed receptors for multiple signals. This suggests that PNECs can function as a signaling hub broadcasting airway sensory information to cells throughout the lung. The richest targets by far were the innervating pulmonary sensory neurons PSN4 and PSN7, which expressed

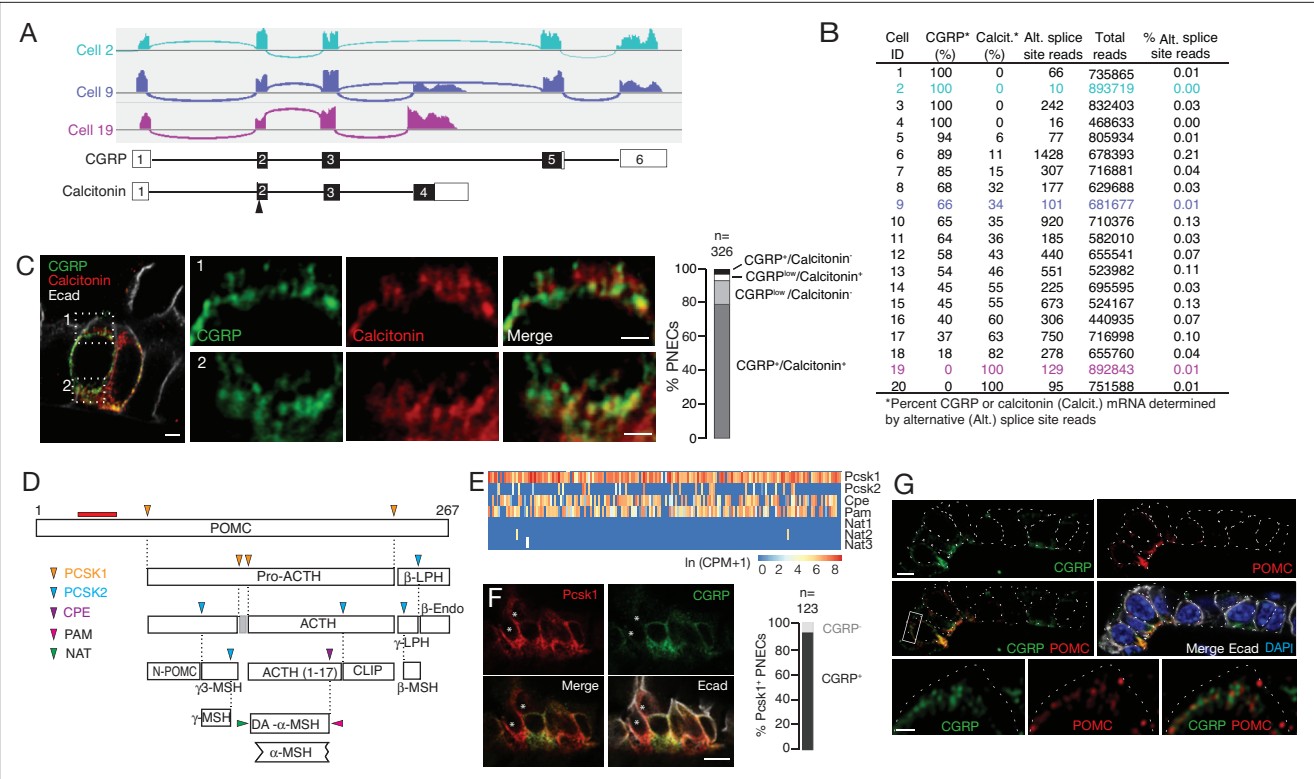

**Figure 3.** Additional PNEC peptidergic diversity from post-transcriptional processing. (**A**) Sashimi plots (top) showing mapped scRNA-seq reads for *Calca* gene and deduced alternative splicing patterns for three representative PNECs (cells 2, 9 and 19 in B), and the resultant mRNAs structures (bottom) encoding either CGRP or calcitonin, with exons numbered (black fill, coding exons). Arrowhead, translation start site. Note cell 2 expresses only CGRP, cell 19 expresses only calcitonin mRNA, and cell 9 expresses both. (**B**) Quantification of alternative splicing of *Calca* mRNA as above for 20 randomly selected *Calca*-expressing PNECs (sashimi plots in *Figure 3—figure supplement 1*). (**C**) Fluorescence super-resolution (AiryScan SR) confocal micrograph (left) of neighboring PNECs in adult PN68 wild type mouse lung immunostained for CGRP (green), calcitonin (red), and E-cadherin (Ecad, white) to show epithelial cell boundaries, with DAPI nuclear counterstain (blue). Close-ups of boxed regions (right) show split and merged channels of apical (1) and basal (2) regions of a PNEC. Most PNECs express both of these peptides (see quantification at right), but note that the neuropeptide (CGRP) and hormone (calcitonin) localize to separate vesicles. Scale bar, 2µm (inset 2µm). Right, quantification of immunostaining (n=326 PNECs scored in 3mice). (**D**) Classical post-translational processing scheme for pro-opiomelanocortin (POMC) in anterior pituitary, showing cleavage sites (arrowheads) of endopeptidases proprotein convertase subtilisin/kexin type 1 (PCSK1, orange) and PCSK2 (blue), and carboxypeptidase E (CPE, purple), and modification sites (arrowheads) of peptidyl-glycine–amidating monooxygenase (PAM amidation site, red) and N-acetyltransferase (NAT acetylation site, green) (*Harno et al., 2018*). ACTH, adrenocorticotrophic hormone; β-LPH, lipotropin, MSH, melanocyte stimulating hormone, Endo, endorphin. Gray box, junctional peptide; red bar, antigen (residues 27–52) of POMC antibody in panel G. In pituitary, other PCSK2 cleavage events produce additional peptides (γ3-MSH, ACTH (1-17), γ-LPH, β-endorphin) although γ−MSH and β-MSH are likely not produced in mouse due to absence of those dibasic cleavage sites. (**E**) Heatmap of expression of POMC processing genes in individual PNECs (n=176) from scRNA-seq dataset. Note expression of *Pcsk1*, *Cpe*, and *Pam* in most (94%, 65%, 74%), *Pcsk2* in some (16%), and *Nat2* and *Nat3* in rare (1%) PNECs; none expressed *Nat1*. This predicts production of all the major pituitary peptides in PNECs, with individual PNECs producing different sets of POMC peptides due to differential expression of the processing enzymes. (**F**) Micrograph of NEB in adult PN90 wild type (CD-1) mouse lung immunostained for PCSK1 (red), CGRP (green), and E-cadherin (Ecad, white). Right, quantification of immunostaining (n=123 PNECs scored in three mice). Most PNECs co-express PCSK1 and CGRP, although rare PNECs (~10%) express only PCSK1 (asterisks in micrograph). Scale bar, 10µm. (**G**) Super-resolution confocal micrograph of NEB in adult PN155 wild type (CD-1) mouse lung immunostained for CGRP (green), POMC (red), and E-cadherin (Ecad, white), with DAPI counterstain (blue). Bottom panel, close-up of boxed region showing separate POMC and CGRP puncta. Quantification (n=347 distinct puncta scored in 8 PNECs co-expressing CGRP and POMC) showed 157 CGRP+ puncta, 190 POMC+ puncta, and no puncta with co-localization of the two. Bar, 10µm (inset 2µm).

The online version of this article includes the following figure supplement(s) for figure 3:

**Figure supplement 1.** *Calca* alternative splicing in mouse PNECs.

receptors for 17 and 19 PNEC peptidergic signals, respectively. Other rich targets included goblet cells (4 signals) and club cells (3 signals). PNECs themselves expressed IGF receptors, implying auto-crine signaling. The results also suggest a broad neuroendocrine-immune signaling axis, with CGRP potentially targeting at least five different types of immune cells (monocytes, dendritic cells, T cells,

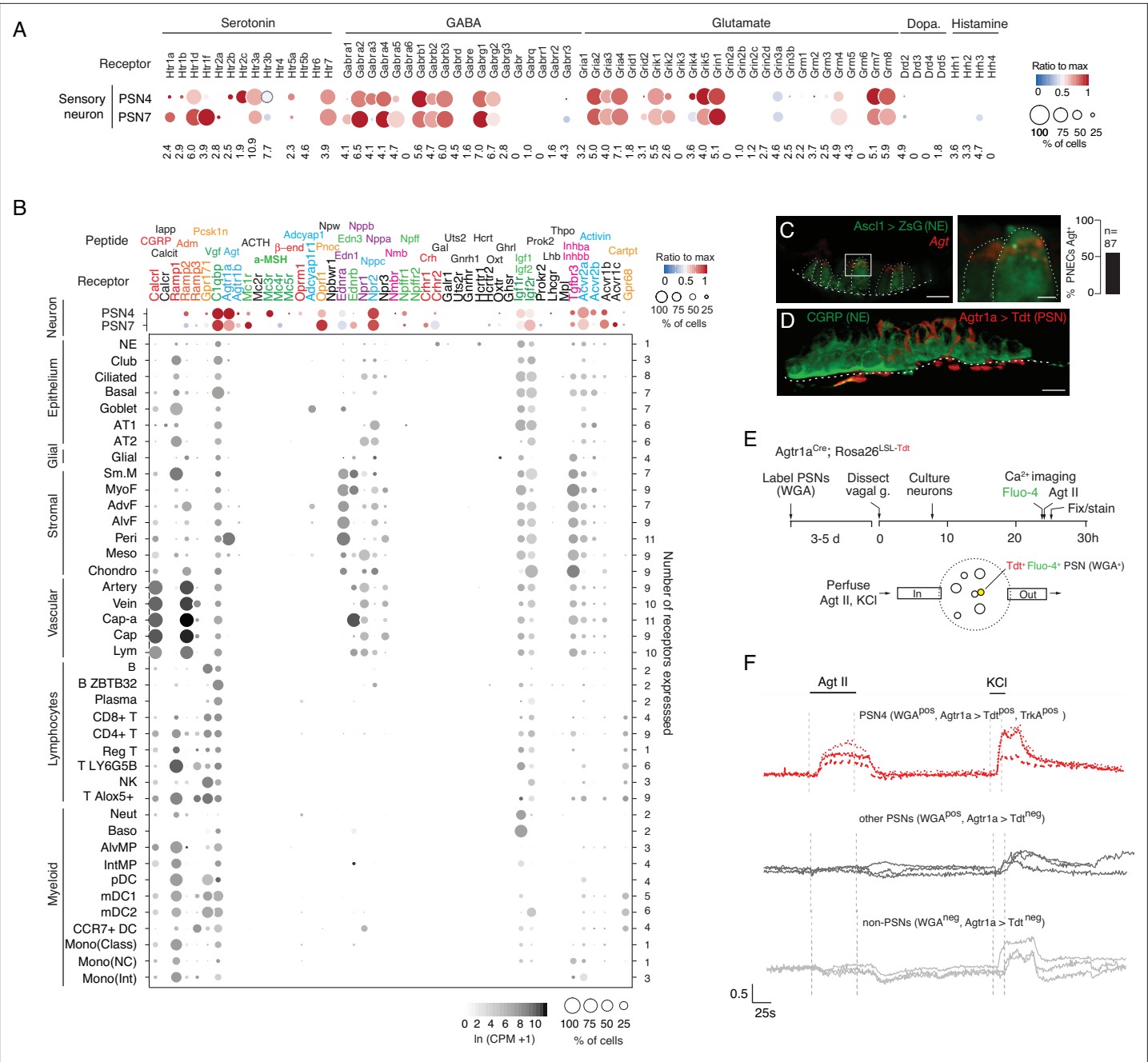

**Figure 4.** Predicted lung cell targets of PNEC signals from expression of their receptors. (**A**) scRNA-seq dot plot of mean expression level (dot heatmap, ratio relative to maximum value for 10 pulmonary sensory neuron (PSN) subtypes, *Liu et al., 2021*) and percent of cells in population with detected expression (dot size) in the NEB-innervating PSN subtypes (PSN4, PSN7) of genes encoding receptors for each of the indicated PNEC neurotransmitters. (**B**) scRNA-seq dot plot of expression in PSN4 and PSN7 (as above) and mean level of expression (dot intensity) and percent of cells in population with detected expression (dot size) in cell types (arranged by tissue compartment) in mouse lung cell atlas *Travaglini et al., 2020* of genes encoding receptors for indicated PNEC peptidergic ligands (shown above corresponding receptor genes). Ligands with expressed receptors are colored; receptors without detectable expression and their corresponding ligands are shown in black. Values (at right), number of PNEC peptidergic signal receptors expressed in at least 10% of the cells of that cell type. CGRP, calcitonin gene-related peptide; Calcit, Calcitonin; Adm, adrenomedullin; Iapp, islet amyloid polypeptide (Amylin); Vgf, neuroendocrine regulatory peptide; Agt, angiotensinogen; α-MSH, alpha melanocyte-stimulating hormone; β-end, β-endorphin; Pnoc, prepronociceptin; Edn1, endothelin-1; Edn3, endothelin-3; Nppc, C-type natriuretic peptide; Nmb, neuromedin B; Npff, neuropeptide FF; Crh, corticotropin releasing hormone; Igf1, insulin-like growth factor 1; Igf2, insulin-like growth factor 2; Inhba, inhibin a; Inhbb, inhibin b; Pcsk1n, proprotein convertase substilisin/kexin type 1 inhibitor; Adcyap1, pituitary adenylate cyclase-activating polypeptide; Npw, neuropeptide W; Nppb natriuretic peptide B; Nppa, natriuretic peptide A; Gal, galanin; Uts2, urotensin II; Gnrh1, gonadotropin-releasing hormone 1; Hcrt, hypocretin;

*Figure 4 continued on next page*

*Figure 4 continued*

Ghrl, ghrelin; Oxt, oxytocin. Cell types: NE, neuroendocrine (PNEC); AT1, alveolar epithelial cell, type 1; AT2, alveolar epithelial cell, type 2; Sm.M, smooth muscle; MyoF, myofibroblast; AdvF, adventitial fibroblast; AlvF, alveolar fibroblast; Peri, pericyte; Meso, mesothelial; Chondro, chondrocyte; Cap-a, capillary aerocyte; Cap, general capillary (Cap-g); Lym, lymphatic cell; B ZBTB32, B cells (ZBTB32+); Reg T, T cells (regulatory); T LY6G5B, T cells (LY6G5B+); NK, natural killer; T Alox5+, T cells (Alox5+); Neut, neutrophil; Baso, basophil; AlvMP, alveolar macrophage; IntMP, interstitial macrophage; pDC, plasmacytoid dendritic; mDC1, myeloid dendritic, type 1; mDC2, myeloid dendritic, type 2; CCR7+ DC, dendritic cell (Ccr7+); Mono(Class), monocyte (classical); Mono(NC), monocyte (non-classical); Mono(Int), monocyte (intermediate). (**C**) Photomicrograph of NEB in adult (PN57) *Ascl1*^CreER; *Rosa26*^LSL-ZsGreen mouse lung with PNECs (NE cells) labeled with Zsgreen (green) and probed by smFISH (PLISH) for *Angiotensinogen* (*Agt*) mRNA (red). Dashed white line, basement membrane. In close up of boxed region (right), note localization of *Agt* mRNA near apical (luminal) surface of PNECs. Scale bars, 10μm (2μm, right panel). Quantification (right) of *Agt*-expressing PNECs (n=87 PNECs scored in two animals). (**D**) NEB of adult (PN90) *Agtr1a*^Cre mouse with Cre-responsive reporter virus (AAV-*flex-tdTomato*) injected into vagal ganglion and co-stained for tdTomato (red, *Agtr1a*-expressing PSNs) and for CGRP (green, PNECs). *Agtr1a*-expressing PSNs penetrate basement membrane (dashed white line) of NEB and terminate at or near its apical surface. Scale bar, 10μm. (**E**) Scheme for isolation and functional imaging of *Agtr1a*-labeled pulmonary sensory neuron (PSN) response to Agt II. (Top panel) Wheat germ agglutinin (WGA) was instilled into trachea of *Agtr1a*^Cre;*Rosa26*^LSL-tdTom mice to retrograde label PSNs and genetically label *Agtr1a*-expressing PSNs (tdTomato) in vagal ganglia. Ganglia were dissected, dissociated into single cells, and cultured overnight before loading with calcium indicator Fluo-4 and addition of Agt II. (Bottom) Diagram of labeled, dissociated vagal ganglion cells on confocal microscope platform with perfusion chamber to flow in Agt II or KCl (positive control). Following imaging, cells were stained for PSN4 marker TRKA. (**F**) Representative neuronal traces showing evoked $Ca^{2+}$ changes measured by Fluo-4 fluorescence (y-axis, fold change in fluorescence relative to baseline, ΔF/F) of three NEB-innervating PSN4 neurons (WGA^pos, *Agtr1a*^Cre-lineage-positive (Agtr1a>Tdt^pos), TRKA^pos; upper traces, red) and six control ganglion cells from same animal (three other PSNs, middle traces, black; three non-PSNs, lower traces, grey) in response to Agt II (0.5–1μM for interval indicated) and to KCl (50mM). The individual (separated) traces of these nine neurons are shown in *Figure 4—figure supplement 2C* along with three representative PSN7 traces, and the results of all viable (KCl responsive) sensory neurons analyzed (n=189, including 100 PSNs) across 11 experiments are provided in the source data for this panel. PSN4 traces, cells 1, 9, 23 of experiment 6, plate a; 'other PSN' traces, cells 11, 25, 27 of experiment 6, plate a; 'non-PSN' traces, cells 9, 10, 11 of experiment 6, plate b. Graph scale, 25s (x-axis), 0.5 Fluo-4 ratio (y-axis).

The online version of this article includes the following source data and figure supplement(s) for figure 4:

**Source data 1.** Corresponds to *Figure 4F*.

**Figure supplement 1.** Mouse lung cell expression of receptors for PNEC neurotransmitters.

**Figure supplement 2.** Mouse lung cell expression of renin-angiotensinogen pathway genes.

and alveolar and interstitial macrophages) plus airway smooth muscle and goblet cells (*Figure 4B*), and other signals targeting NK cells, B cells, T cells, and dendritic cells (PCSK1N), basophils and neutrophils (IGFs), and dendritic cells (adrenomedullin).

We experimentally validated one of the inferred signaling interactions—the predicted angiotensin signal from PNECs to pulmonary sensory neurons. Angiotensin is among the most medically important hormones because of its key role in vasoconstriction and blood pressure regulation (*Figure 4—figure supplement 2A*), and because one of its processing enzymes (angiotensin converting enzyme 2, ACE2) also serves as the entry receptor for SARS and Covid-19 coronaviruses. Indeed, the lung plays an essential role in this hormone pathway by providing angiotensin converting enzyme (ACE), the target of a ubiquitous class of anti-hypertensive drugs (ACE inhibitors), that proteolytically processes circulating angiotensin I peptide (Agt I) into the potent vasoconstrictor Agt II. Our discovery that PNECs express angiotensinogen (*Agt*), the preprohormone for Agt II, reveals a pulmonary source of the hormone, and our molecular cell atlas points to three potential lung targets: pericytes, and the PSN4 and PSN7 pulmonary sensory neurons that innervate NEBs, each of which selectively expressed Agt II receptor gene *Agtr1a* (*Figure 4B*).

We confirmed *Agt* expression in PNECs by smFISH, which showed expressing cells localized within NEBs (*Figure 4C*). We also confirmed expression of its receptor *Agtr1a* in NEB-innervating sensory neurons by injecting *Agtr1a*^2A-Cre mice (*Leib et al., 2017*) with a Cre-responsive reporter virus (AAV-*flex-tdTomato*) in the vagal nodose ganglion where cell bodies of pulmonary sensory neurons reside, and found their tdTomato-labeled termini ramifying on NEBs at airway branchpoints (*Figure 4D*). (PNECs within NEBs are the only lung cells known to directly contact *Agtr1a*-expressing pulmonary sensory neuron termini *Liu et al., 2021*). To determine if angiotensin can activate pulmonary sensory neurons, we labeled pulmonary sensory neurons by introducing a fluorescently-labeled wheat germ agglutinin (WGA-A647) into the lung, and allowing 4–5 days for label uptake at sensory neuron termini and retrograde transport to their cell bodies in the vagal ganglia (*Figure 4E*). We then isolated and cultured vagal ganglion cells and visualized their neuronal activity by Fluo-4 calcium imaging during perfusion of Agt II into the imaging chamber. Agt II activated 1–15% of cultured pulmonary sensory neurons,

as well as rare sensory neurons from other organs (*Figure 4F*). We identified responding pulmonary sensory neurons as the PSN4 subtype: they expressed *Agtr1a* $^{2A-Cre}$ lineage-label and stained positive for PSN4-specific marker neurotrophic receptor tyrosine kinase 1 (TRKA). PSN7 (*Calb1+*) neurons, which are larger but express *Agtr1a* at lower levels and do not express *Ntrk1*(TrkA) (*Liu et al., 2021*), were not activated under these conditions (*Figure 4—figure supplement 2C*). The lower levels of *Agtr1a* expression may explain the selective activation of PSN4 neurons, although other explanations are possible including that PSN7 neurons are less healthy under our assay conditions. The results suggest that in addition to its classical role as a circulating vasopressor, Agt II can function as a local neuromodulator from PNECs directly to PSN4 sensory neurons, transmitting airway sensory information to the brain.

For one-third (36%) of the PNEC peptidergic signals (calcitonin/*Calca*, amylin/*Iapp*, ACTH/*Pomc*, neuropeptide W/*Npw*, galanin/*Gal*, urotensin 2/*Uts2*, gonadotropin-releasing hormone/*Gnrh1*, hypocretin/*Hcrt*, ghrelin/*Grhl, oxytocin/Oxt, prokinectin 2/Prok2, leutinizing hormone, subunit B/Lhb,thrombopoietin/Thpo*), we did not detect appreciable expression of their receptors in any lung cell type or PSN (*Figure 4B*). All of these classically function as circulating hormones with targets throughout the body, such as calcitonin (bones, kidneys), ACTH (adrenal cortex, adipocytes), amylin (brain stem), and inhibin (pituitary), so PNECs too may secrete these hormones into circulation. However, we cannot exclude that some have local targets but their receptors were expressed below detection or in rare or fragile cells not captured in our lung cell atlas.

## PNECs are diverse, multimodal sensors

Classical physiological studies of PNECs indicate that signal secretion is triggered by a variety of stimuli including hypoxia, hypercapnia, mechanical stimuli, and allergens (*Lembrechts et al., 2012*; *Livermore et al., 2015*; *Sui et al., 2018*; *Youngson et al., 1993*). However, the full diversity of PNEC sensory functions are unknown, and the molecules that mediate these functions have only recently begun to be identified (*Nonomura et al., 2017*). To more fully elucidate PNEC sensory functions and the molecules that mediate them, and to determine how sensors are paired with the myriad PNEC signals described above, we curated a list of over 1500 mouse genes encoding extant mammalian sensory receptors and their homologues (*Supplementary file 6*) including ones previously implicated in PNEC sensory function, then searched our scRNA-seq dataset for ones selectively expressed in PNECs.

The mechanically activated channel PIEZO2 (*Fam38b*) gene was specifically and broadly (>90%) expressed by PNECs (*Figure 5A*, *Figure 5—figure supplement 1A*), as previously described (*Nonomura et al., 2017*). In contrast, PIEZO1 (*Fam38a*) was expressed across all major lung compartments but excluded from PNECs (*Figure 5—figure supplement 1A*). Mechanically activated two-pore potassium channel TREK-2 (*Kcnk10*) gene (*Figure 5A*, *Figure 5—figure supplement 1B*; *Bang et al., 2000*) and family member *Kcnk16* (*Figure 5—figure supplement 1B*) were selectively expressed in PNECs, whereas TREK-1 (*Kcnk2*) was expressed in other airway epithelial cells and almost completely excluded from PNECs (*Figure 5—figure supplement 1B*). Three TRP family cation channel genes also showed selective or enriched expression in PNECs: auditory hair cell stereocilium channel *Trpml3* (*Mcoln3*), *Trpc4*, and *Trpv2*, a noxious heat sensor also implicated in mechano- and osmoregulation (*Figure 5A*, *Figure 5—figure supplement 1A*). *Casr*, a G-protein-coupled receptor implicated in ciliary mechanosensing and previously proposed to integrate NEB signals (*Lembrechts et al., 2013*), was also selectively expressed in PNECs (*Figure 5A*, *Figure 5—figure supplement 1A*), as was *Lhfp15*, an integral membrane protein of the auditory hair cell mechanotransduction complex. These results support the role of PNECs as specialized airway mechanosensors and suggest potentially new mechano- or thermosensory functions mediated by proteins encoded by *Kcnk10*, *Kcnk16*, *Trpml3* (*Mcoln3*), *Trpc4*, *Trpv2*, and *Lhfp15*.

PNECs are proposed to function in $CO_2$ sensing because they can be activated by hypercapnic challenge (*Lauweryns et al., 1977*), and by bicarbonate and acid in vitro (*Ebina et al., 1997*; *Livermore et al., 2015*), a response dependent on carbonic anhydrase, but the proteins that mediate this function are unknown. PNECs selectively expressed the classic acid-sensing potassium channel TREK (*Kcnk3*), and rare PNECs expressed acid-sensing sodium channels ASIC3 (*Accn3*) and ASIC4 (*Accn4*) (*Figure 5A*, *Figure 5—figure supplement 1A*). Expression of the widely-distributed cytoplasmic carbonic anhydrase *Car2* gene was not detected in PNECs, but some expressed membrane-bound

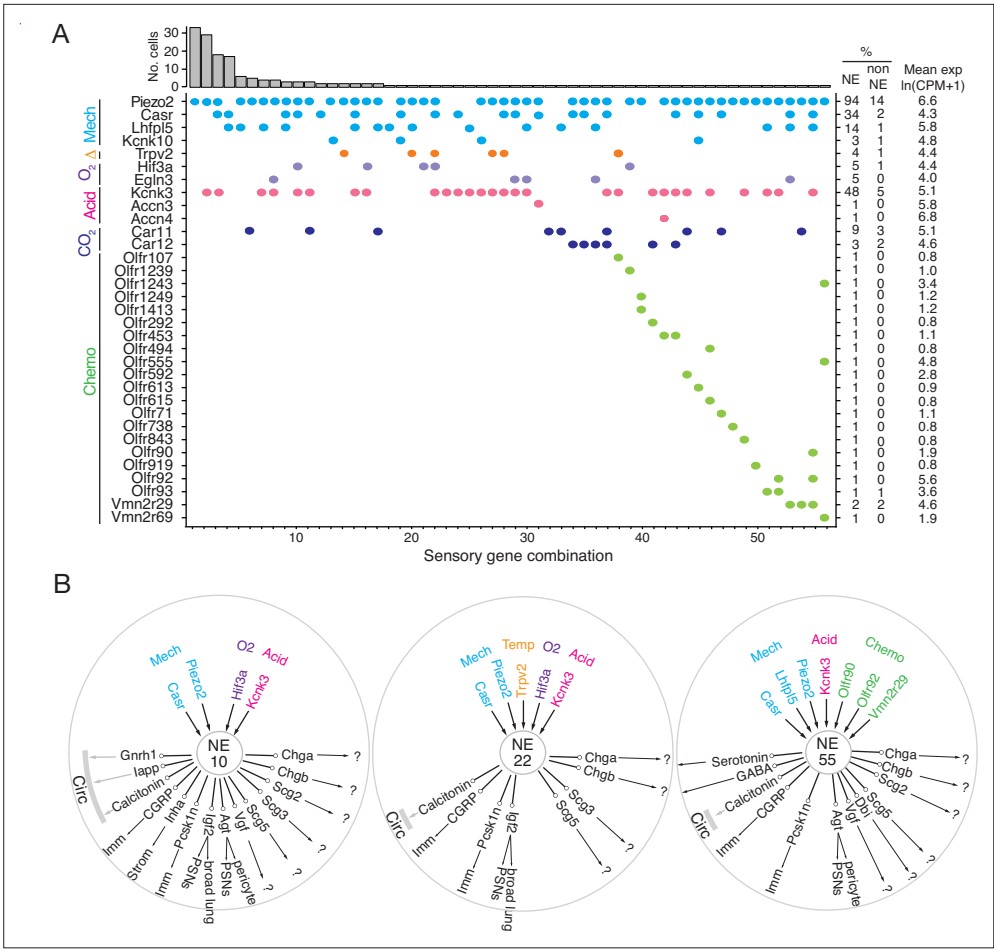

**Figure 5.** Patterns of sensory gene expression in individual PNECs. (**A**) Patterns of expression from scRNA-seq profiles of sensory genes (colored dots, expressed genes) in individual PNECs (n=176). Histogram shows number of PNECs observed with the pattern of sensory gene expression below it. Values at right are percent of PNECs (NE) and other (non-NE) profiled cells that express the gene and the mean expression level for expressing PNECs. Sensory genes are grouped by modality: mechanosensing (Mech); thermosensing ($\Delta$); oxygen sensing ($O_2$); acid-sensing (Acid); $CO_2$-sensing, carbonic anhydrases ($CO_2$), chemosensing (Chemo). Note the 56 different combinations of expressed sensory genes, with most PNECs predicted to be multimodal because they express more than one class of sensor. (**B**) Schematics of sensor and signal genes expressed by three individual PNECs. Number in center indicates sensory gene combination in panel A. Genes above number are expressed sensory genes, and genes below number are expressed signal genes with arrows indicating lung targets (cells that express receptor) or signals without identified lung targets that may enter circulation (circ) to target other organs.?, signals without known receptors. Imm, multiple immune populations; PSN, pulmonary sensory neurons.

The online version of this article includes the following figure supplement(s) for figure 5:

**Figure supplement 1.** Mouse lung cell expression of sensory genes.

**Figure supplement 2.** Mouse lung cell expression of TNF family ligand and receptor genes.

*Car12* and the carbonic anhydrase-related gene *Car11* (**Figure 5A**, **Figure 5—figure supplement 1A**).

One of the first and still the most prominent proposed function of PNECs is as airway oxygen sensors because they can be activated by hypoxic challenge in vivo (**Lauweryns et al., 1978**) and in cultured lung slices or as isolated PNECs (**Youngson et al., 1993**). The oxygen sensing mechanism is still uncertain but the dominant hypothesis proposes that low oxygen reduces $H_2O_2$ generation by a membrane-bound NADPH oxidase (heterodimer of gp91phox/*Cybb* and p22phox/*Cyba*, plus regulatory subunits p47phox/*Ncf1*, p67phox/*Ncf2*), which inhibits an oxygen-sensitive potassium channel (Kv3.3/*Kcnc3* and Kv4.3/*Kcnd3*) that activates L-type voltage-gated calcium channels (**Fu**

*et al., 2002*; *Youngson et al., 1993*), triggering neurosecretion that may act locally or be propagated to the brainstem breathing center. We did not detect PNEC expression of NADPH oxidase subunits p91phox/*Cybb*, p47-phox/*Ncf1*, or p67-phox/*Ncf2*, and p22phox/*Cyba* was broadly expressed in all cells (*Figure 5—figure supplement 1A*). Kv3.3/*Kcnc3* and Kv4.3/*Kcnd3* were detected only at low levels or in rare cells (previous reports of Kv3.3/*Kcnc3 and* Kv4.3/*Kcnd3* expression by PNECs were based on in situ hybridization studies of fetal rabbit and neonatal human lung (*Gonzalez et al., 2009*; *Wang et al., 1996*), so there may be developmental or species-specific differences in its PNEC expression). Although we did not detect substantial expression of these previously implicated channels, we found robust and selective PNEC expression of many other potassium channel genes (*Figure 5—figure supplement 1B*; e.g. voltage-gated: *Kcnc1, Kcnc2, Kcnb1, Kcnv1, Kcnf1, Kcnq2, Kcnq5, Kcnh2, Kcnh6, Kcnh7, Kcnh8*; cyclic-nucleotide gated Na/K channel: *Hcn1, Hcn2, Hcn3, Hcn4*; calcium-activated: *Kcnn3*; 2-pore: *Kcnk1, Kcnk3*) that could contribute to the hypoxia-sensitive potassium current required for PNEC secretion. We also did not detect selective expression in PNECs of any of the genes (mitochondrial respiratory complexes) implicated in the mitochondrial hypothesis of oxygen sensing (*Mulligan et al., 1981*; *Quintana et al., 2012*; *Stettner et al., 2011*; *Figure 5—figure supplement 1A*). Thus, the identity of the acute oxygen sensor in PNECs remains uncertain, although several of the newly identified PNEC potassium channels are appealing candidates. Chronic hypoxia also influences PNECs, and hypoxia inducible factor *Hif1a* is expressed though not selectively in PNECs, whereas *Hif3a* is a PNEC-selective family member (*Figure 5—figure supplement 1A*).

Primate PNECs have been proposed as volatile chemical sensors based on expression of olfactory receptors OR2W1 and OR2F1 in some solitary human PNECs, and the response of PNECs in tracheobronchial cultures to nonanal and other chemicals (*Gu et al., 2014*). We identified 19 olfactory and two pheromone receptor superfamily genes expressed in rare PNECs (*Figure 5A*, *Figure 5—figure supplement 1A*). We curiously also detected expression of photoreceptor opsin *Opn1sw* and non-visual opsin *Opn3* (*Figure 5—figure supplement 1C*) as well as two TNF receptor family genes (*Tnfrsf12a, Tnfrsf21*; *Figure 5—figure supplement 2A*), suggesting possible light and immune sensing functions for PNECs.

Individual PNECs express multiple sensors and are predicted to sense multiple modalities. For example, one PNEC (combination 55) expressed mechanoreceptor/transducer genes *Piezo2, Casr*, and *Lhfpl5*, acid-sensitive channel *Kcnk3*, and chemoreceptors *Olfr90*, *Olfr92*, and *Vmn2r29* (*Figure 5A and B*). Individual PNECs expressed different combinations of sensory genes, indicating diversity in their sensory roles (*Figure 5A and B*). Comparison of the sensors and signals expressed in each PNEC did not identify any strong correlations, suggesting that specific sensory inputs are coupled to different output signals in different PNECs.

## Human PNECs also show diverse sensory, signaling and target profiles

To explore the generality and biomedical significance of the properties of mouse PNECs uncovered by scRNA-seq, we performed a similar analysis of human PNECs. Although human PNECs are also extremely rare, in our scRNA-seq study of ~75,000 human lung cells (*Travaglini et al., 2020*) we obtained expression profiles of 55 PNECs. We analyzed these PNEC profiles as we did above for mouse PNECs and found that, even with this more limited sample, all the features uncovered for mouse PNECs are also apparent for human PNECs, although in more extreme form for some features and with species-specific specializations.

Human PNEC markers are largely conserved with mouse and include 26 that are more sensitive (e.g. *GRP, SCGN, SCG5, BEX1*), specific (*SLC35D3, CPLX2*), and/or highly expressed (*SCG2*) than the four common clinical markers (*CHGA, SYP, INSM1, ASCL1*; *Figure 6—figure supplement 1*). Some of the best markers are species-specific (human-specific: *GRP, SCGN*; mouse-specific: *Resp18*), and transcript isoform mapping of the human ortholog (*CALCA*) of the classic mouse PNEC marker CGRP (*Calca*) revealed alternative splicing such that only 60% of human PNECs expressed CGRP isoforms whereas all expressed calcitonin isoforms (*Figure 6—figure supplement 2*), explaining why calcitonin but not CGRP is a good human PNEC marker (*Weichselbaum et al., 2005*) whereas both are valuable for mouse (*Figures 1B and 3C*, *Figure 3—figure supplement 1*).

Like mouse, human PNECs have a large and diverse signaling output. Human PNECs express biosynthetic genes for neurotransmitters serotonin (*TPH1*) and GABA (*GAD1*) (*Supplementary file 8*), the major neurotransmitters of mouse PNECs. Some human PNECs are also likely glutamatergic

because 14% expressed glutamate vesicular transporter *SLC17A6*, and some may be catecholaminergic or glycinergic because rare PNECs expressed key catecholamine synthetic enzymes (*DBH*, *PNMT*) or glycine re-uptake transporter *SLC6A5* (*Supplementary file 8*). Expression of dopaminergic genes was detected in rare mouse PNECs but none of the analyzed human PNECs.

Human PNECs expressed 40 different peptidergic genes (*Figure 6A*), 45% of the 93 annotated human peptidergic genes (*Supplementary file 4*), with individual PNECs expressing 12.2±2.7 (mean ± SD, range 6–18), twice as many as mouse PNECs (*Figure 6A*). Like mouse, almost every PNEC (54 of 55 cells, 98%) expressed a different combination (*Figure 6B*). Two-thirds (26/40, 65%) of the expressed peptidergic genes are also expressed in mouse PNECs, although *CARTPT* was expressed in many fewer human PNECs (4% vs 18% in mouse) and *POMC* in many more (57% vs 4% in mouse) (*Figure 6A*, *Supplementary file 4*). Remarkably, 13 of the 14 human-specific peptidergic genes encode hormones (*Figure 6A*, *Supplementary file 4*), including some of the most biomedically significant: erythropoietin/*EPO*, renin/*REN*, five hypthothalamic releasing/inhibitory hormones and pituitary regulators (thyrotropin-releasing hormone/*TRH*, prolactin-releasing hormone/*PRLH*, gonadotropin-releasing hormone 2/*GNRH2*, corticotropin releasing hormone urocortin/*UCN* and inhibitory hormone somatostatin/*SST*), the common subunit of multiple pituitary hormones (glycoprotein hormone subunit A/*CGA*), reproductive organ developmental regulator anti-mullerian hormone/*AMH*, digestive hormones gastrin-releasing peptide/*GRP* and cholecystokinin/*CCK*, and the potent vasoregulators urotensin 2B/*UTS2B* and kininogen/*KNG1*. Of the 17 mouse-specific PNEC peptidergic genes (*Supplementary file 4*), all except three (*Igf2*, 72% of PNECs; *Ucn2*, 13%; *Iapp*, 7%) were detected only in rare PNECs (1–3%) so could also be rare in human PNECs and found on further profiling.

The human lung cell expression patterns of receptors for the 32 PNEC peptidergic signals with known receptors are shown in *Figure 6—figure supplement 3A*, identifying potential direct targets in lung. As for mouse, expression patterns were diverse and almost all lung cell types expressed receptors for one or more signals, implying human PNECs can also transmit pulmonary sensory information throughout the lung. The predicted targets of the conserved PNEC signals were also largely conserved, for example broad stromal and vascular targeting by inhibin, immune cell targeting by CGRP and VGF, and pericyte targeting by angiotensin. Autocrine signaling appears prominent, as PNECs express receptors for ghrelin and erythropoietin, one of the human-specific signals, and almost every PNEC neurotransmitter (GABA, glutamate, dopamine, and epinephrine/norepinephrine; *Figure 6—figure supplement 3B*). Curiously, the sole PNEC autocrine signal identified in mouse, IGF2, was not detected in any profiled human PNEC (*Figure 6A*, *Supplementary file 4*). No receptor expression was detected in lung for nearly half (45%, 15 of 33) the human PNEC peptidergic signals, including 10 of the 14 human-specific signals. While some of these signals may target pulmonary sensory neurons or rare pulmonary cells not captured in our human lung atlas, others may enter circulation and target sites beyond the lung.

Expression of sensory genes (*Figure 6—figure supplement 4*, *Figure 6—figure supplement 5*) indicates that, as in mouse, human PNECs are multimodal sensors with almost all cells expressing different combinations of sensor genes for diverse stimuli. These include orthologues of mouse PNEC sensors such as mechanically-activated channels *PIEZO2* and *KCNK10*, thermosensor *TRPV1*, carbonic anhydrase *CA11*, and acid-sensitive channel TASK-1/*KCNK3*, plus human-specific TASK-3/*KCNK9* with a more acidic range (pK 6.0–6.7; *Duprat et al., 1997*). Like mouse, human PNECs express 19 different olfactory receptor (OLFR) genes but in a greater proportion of cells (33% vs 11% for mouse), with individual cells expressing up to five different OLFR genes. As in mouse, NADPH-oxidase complex genes proposed as PNEC oxygen sensors were not detected or not specifically expressed by PNECs (*Figure 6—figure supplement 5*), but two human PNEC OLFR genes (*OR51E1*/*Olfr558*, the most widely-expressed, in 16% of PNECs, and *OR51E2*/*Olfr78*) are a close family member and orthologue of OLFR78, which is activated by lactate and implicated in acute hypoxia-sensing in the mouse carotid body (*Chang et al., 2015*). *OR51E1* and *OR51E2* were co-expressed along with acid-sensitive channel *KCNK3* in a single human PNEC (*Figure 6—figure supplement 4A*, combination 25), so this cell may be specialized for hypoxia or chemosensing. As in mice, rare human PNECs also expressed a pheromone receptor (*VN1R1*) and opsins (*OPN1SW*, *OPN3*), but human PNECs also expressed bitter (*TAS2R10, TAS2R31, TAS2R5*) and sour taste receptors (*PKD2L1, PKD1L3*) and the trace amino acid receptor *TAAR1* (*Figure 6—figure supplement 4A*).

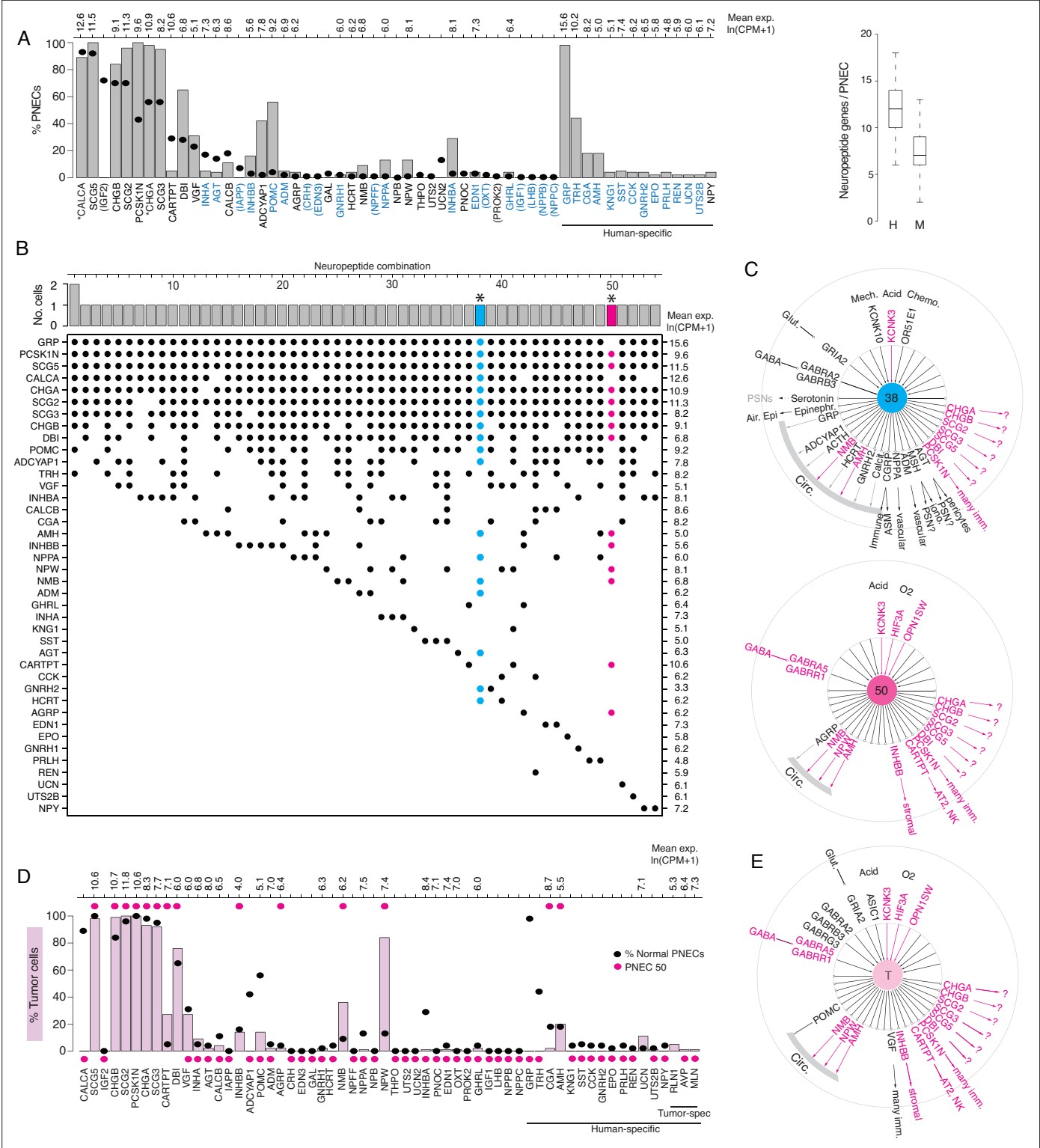

**Figure 6.** Expression of sensory and signaling genes in normal human PNECs and a pulmonary carcinoid. (**A**) Histogram of peptidergic gene expression in normal human PNECs (n=55 PNECs from 2subjects) by scRNA-seq. Bars show percent of PNECs with detected expression of the indicated peptidergic gene; values above bars are mean gene expression (ln (CPM + 1)) in expressing cells. Peptidergic genes are listed in same order as orthologous mouse genes in ***Figure 1E***, and ellipses show percent of mouse PNECs that express the orthologous genes (from ***Figure 1E***, ***Supplementary file 4***) for comparison. Among the 41 total peptidergic genes detected in human PNECs, the 14 genes at right ("human-specific") were expressed in human but not mouse PNECs; genes in parentheses were expressed only in mouse PNECs. *, previously known human PNEC peptidergic

*Figure 6 continued on next page*

*Figure 6 continued*

genes; blue font, classic hormone genes. Box and whisker plots (right) show quantification of peptidergic genes expressed per cell in human (**H**) and mouse (**M**) PNECs. Human values: 12.2±2.7 (mean ± SD; range 6–18, median 12, mode 12) expressed peptidergic genes. (**B**) Combinations of peptidergic genes expressed in individual human PNECs (n=55) from scRNA-seq. Histogram (top) shows number of PNECs expressing each of the 54 observed combinations of expressed peptidergic genes (dots, bottom). Values at right are mean expression (ln (CPM + 1)) of indicated gene in expressing cells. *, individual PNECs diagrammed in panel C (#38, blue; #50, magenta). (**C**) Schematics (as in *Figure 5*) of sensor and signal genes expressed by individual human PNECs (combination #38, cell ID F21_B001235; #50, cell ID D5_B003269). Sensory and secretory genes expressed by cell #50 and by either cell #38 or the composite tumor cell (in panel E) are highlighted in magenta. (**D**) Histogram as in panel A of peptidergic gene expression in PNEC-like tumor cells of a human bronchial carcinoid (n=330 profiled tumor cells from one subject) from scRNA-seq. Pink bars show percent of tumor cells with detected expression of the indicated peptidergic gene; values above bars are mean gene expression (ln (CPM + 1)) in expressing cells. To facilitate comparison, the percent of normal human PNECs that express each peptidergic gene (black ellipses, from panel A) and peptidergic gene expression in the single normal PNEC that most closely matches the tumor cell profile (magenta ellipses, PNEC #50 in panels B), (**C**) are included in plot. Tumor-specific (spec), peptidergic genes expressed in tumor but not normal PNECs. (**E**) Schematic (as in panel C) of sensor and signal genes expressed by a 'composite' carcinoid tumor cell (**T**) from data in D. Genes shown are those expressed in>15% of the profiled tumor cells, plus *POMC* (14% of tumor cells), *INHBB* (14%), and *OPNSW1* (6%). Genes expressed by both 'composite' tumor cell and normal PNEC #50 (panel C) are highlighted in magenta.

The online version of this article includes the following figure supplement(s) for figure 6:

**Figure supplement 1.** Human PNEC markers.

**Figure supplement 2.** Alternative splicing of CALCA transcripts in human PNECs.

**Figure supplement 3.** Human lung cell expression of receptors for human PNEC neuropeptides, peptide hormones, and neurotransmitters.

**Figure supplement 4.** Patterns of sensory gene expression in human PNECs.

**Figure supplement 5.** Human lung cell expression of sensory genes and potassium and calcium channel genes.

**Figure supplement 6.** Expression of peptidergic genes in human lung carcinoids profiled by bulk RNA-seq.

Thus, as in mouse, almost every human PNEC is equipped to perceive multiple diverse stimuli and expresses a large number of peptidergic signals and neurotransmitters that can be received by a variety of cells within the lung and potentially throughout the body, like the signals from PNEC-derived tumors described below.

## scRNA-seq profile of a human lung carcinoid: amplification of a rare PNEC

PNECs are the presumed origin of a variety of human lung neuroendocrine tumors (*Travis et al., 2015*) that can cause diverse ectopic hormone syndromes such as classic carcinoid syndrome (wheezing, flushing, diarrhea, increased heart rate), Cushing's syndrome (*Arioglu et al., 1998*), and acromegaly (*Athanassiadi et al., 2004*). To determine the full sensory and signaling potential of a PNEC tumor and their relationship to those of normal PNECs, we obtained scRNA-seq profiles of NE tumor cells of a lung carcinoid from a 51-year-old female patient with onset of idiopathic hypertension in the year preceding therapeutic lung resection. The profiled tumor cells expressed most general PNEC markers (e.g. *SCG5, CHGB, SCG2, PCSK1N, CHGA, SCG3*) indicating retention of PNEC identity. However, they did not recapitulate the full spectrum of PNEC diversity in their sensory and signaling gene profiles (*Figure 6D and E*). Some peptidergic genes such as *CALCA* and *GRP* that are expressed in almost all normal PNECs were expressed in few if any tumor cells, whereas *NPW, NMB*, and *CARTPT* that are expressed in only rare PNECs were expressed in most (*NPW*) or many (*NMB, CARTPT*) of the tumor cells (*Figure 6D*). Likewise, the tumor cells lacked expression of the common PNEC mechanosensor *PIEZO2* but showed broad expression of the acid-sensitive channel *KCNK3* and opsin *OPN1SW*, which is expressed only by rare normal PNECs (*Figure 6—figure supplement 4*). This suggests that proliferating tumor cells retain a "memory" (albeit imperfect) of the expression profile of the PNEC from which they originated. And, because this tumor's unusual expression profile nearly matched that of a rare normal PNEC (#50, *Figure 6B and C*), which like the tumor cells expressed *NPW, NMB, CARTPT, KCNK3*, and *OPN1SW* and lacked expression of common PNEC genes *CALCA, GRP*, and *PIEZO2* (*Figure 6D and E*, *Figure 6—figure supplement 4*), we suggest that this carcinoid arose by transformation of a cell similar to PNEC #50. Tumor cell expression of *NPW* may have caused the patient's hypertension because the active peptide (NPW-23) is a circulating hormone that regulates vascular tone and has been proposed to play a role in the pathophysiology of hypertension (*Ji et al., 2015*; *Yu et al., 2007*).

To determine the peptidergic signaling profiles of other lung carcinoids, we analyzed peptidergic gene expression in 111 human lung carcinoids profiled by bulk RNA-seq (*Alcala et al., 2019*). This revealed prominent expression of 70% (28 of 40) of the normal human PNEC peptidergic genes, including new genes we identified in rare subpopulations of normal PNECs, in at least some of the tumors (*Figure 6—figure supplement 6*). Ten peptidergic genes expressed by lung carcinoids, including four previously detected in clinical samples (GHRH, PENK, TAC1, VIP), were not found in our normal PNECs; however, all of these were expressed in only rare carcinoids so may be expressed in rare PNECs and if so should be identified on further PNEC profiling. Interestingly, some of the peptidergic genes expressed in only minor subpopulations of normal PNECs (e.g. *NPW, NPPA, SST, CARTPT*) were detected in many carcinoids, whereas the nearly ubiquitous PNEC peptidergic gene *CALCA* was absent from most carcinoids. This suggests that not all normal PNECs are equally susceptible to carcinoid transformation.

## Discussion

By single-cell expression profiling of hundreds of the exceedingly rare PNECs in mouse and human, we discovered they express over 40 peptidergic genes, nearly half of all such genes including many classic hormones. Individual PNECs express up to 18 peptidergic genes, with almost every cell expressing a distinct combination. The diversity of expressed signals is further increased by alternative splicing, and by post-translational processing as inferred from expression of prohormone processing genes. These diverse signals can directly target a wide array of cell types in the lung, predicted by expression of the cognate receptors, including almost every cell type across all five tissue compartments: epithelial (including putative autocrine PNEC signals), endothelial, stromal, immune, and neural. The richest targets are pulmonary sensory neurons (PSNs) that innervate PNECs. We confirmed one predicted signal to PSNs, angiotensin (Agt II, mature product of *Agt*), can directly activate a recently identified PSN subtype (PSN4) that innervates PNECs, expresses its receptor (*Agtr1a*), and projects to the brainstem to regulate respiratory rate *Liu et al., 2021*; Diaz de Arce, et al., unpublished data. Hence in addition to its classical role as a circulating vasopressor whose receptor is targeted by major anti-hypertensive drugs, Agt II may serve as a neuromodulator in the breathing circuit. Eighteen other PNEC signals are also classical hormones, but unlike Agt II their receptors are not expressed in any cell types in the lung cell atlases, suggesting that PNECs could contribute to the circulating pool of these hormones. PNECs are thus extraordinarily rich and diverse signaling hubs that produce scores of neuropeptides and peptide hormones that can signal directly to many cells in the lung, to the brain through pulmonary sensory neurons, and potentially to cells throughout the body through the circulation.

PNECs are scattered throughout the airway epithelium and form large clusters at bronchial branch-points, so they are ideally positioned to serve as sentinels that monitor inhaled air and airway status. Our single-cell data imply that each PNEC is in fact a multimodal sensor, expressing a distinct combination of mechanical, thermal, and acid sensors along with carbonic anhydrases important in $CO_2$ sensing, diverse chemosensors including olfactory receptors, vomeronasal receptors and taste receptors, and even light-sensing opsins. Different combinations of sensors are co-expressed in individual PNECs along with different combinations of signals (*Figures 5B and 6C*). Because the peptidergic signals are packaged in separate vesicles, secretion of each signal could be independently regulated, presumably in response to activation of a different sensor (or sensor combination). In this way PNECs can simultaneously monitor many aspects of airway status at many positions along the airway, and selectively transmit this information to target cells in the lung, the brain, and the rest of the body.

In many ways PNECs resemble enteroendocrine cells (EECs), the sentinels scattered along the gut epithelium to monitor nutrients, microbial products, and other luminal contents and signal that information locally in the gut to coordinate ingestion, absorption, metabolism, and disposal, and throughout the body and brain to regulate mood and appetite (*Bai et al., 2019*; *Bellono et al., 2017*; *Kaelberer et al., 2018*). The enteroendocrine system is commonly called the 'gut-brain axis' and is considered the largest endocrine organ because of its many endocrine cells and signals. Our data suggest it may be rivaled or even surpassed by the pulmonary neuroendocrine system, which expresses more than double the ~20 signals produced by EECs (*Beumer et al., 2020*). Each EEC apparently expresses only one or a few peptidergic signals and neurotransmitters, which define 12 classical EEC subtypes (*Worthington et al., 2018*), whereas individual PNECs express 5–10 times

more and their expression patterns define at least an order of magnitude more molecular subtypes. While PNECs have long been speculated to serve as local signaling centers in the lung and fast conduits of sensory information to the brain through afferent sensory neurons, our data suggest that like EECs they also serve a more global endocrine function. This would explain why the basal surface of some PNECs, where secretory vesicles are densely packed, are apposed to fenestrated capillaries (*Lauweryns et al., 1973*). Although PNEC contribution to circulating hormone pools under normal physiological conditions is yet to be demonstrated, they contribute at least under pathological conditions. Indeed, we found that the extensive PNEC signaling repertoire described here including most of newly identified peptidergic genes, were collectively expressed in the 111 available cases of profiled human lung carcinoids. An individual human carcinoid, however, expresses a discrete set of signals resembling that of a normal PNEC, suggesting that each such tumor amplifies the set of signals expressed by the tumor-initiating PNEC, thereby explaining the diversity of carcinoid syndromes (*Limper et al., 1992*; *Pernow and Waldenström, 1957*; *Shalet et al., 1979*). PNECs may comprise a second global signaling axis we dub 'the lung-brain axis'.

This new understanding of PNEC function and their extraordinary diversity, including many sensor and signaling genes detected in only a single profiled cell, required pre-enrichment (mouse) or massive profiling (human) to obtain just the first few hundred expression profiles of these exceedingly rare cells. The neuroendocrinology of the lung our study reveals has broad implications for medicine even beyond NE cell tumors (*Rudin et al., 2019*; *Travis et al., 2015*; *Young et al., 2011*), including the many other pulmonary diseases such as asthma (*Sui et al., 2018*), SIDS (*Cutz et al., 2007*; *Mou et al., 2021*), and bronchopulmonary dysplasia (*Gillan and Cutz, 1993*) that have been associated with PNEC abnormalities and perhaps now extending to diseases outside the lung. The results already have implications for the Covid-19 pandemic because SARS-CoV-2 virions use an angiotensin pathway regulator (ACE2) to enter and destroy lung cells and cripple gas exchange along with the ability of the patient to sense the deficit. Although expression of angiotensinogen and many other PNEC sensory and signaling genes are conserved so their functions can be explored in mice, we also uncovered 13 human-specific PNEC signaling genes encoding classical hormones (e.g. ACTH (*POMC*), GRP, TRH, AMH, CCK). Hence the lung-brain axis may be especially prominent in humans.

# Materials and methods

**Key resources table**

| Reagent type (species) or resource | Designation | Source or reference | Identifiers | Additional information |
|---|---|---|---|---|
| Antibody | Anti-PC1/Pcsk1 (rabbit polyclonal) | Abcam | Cat# ab3532, RRID:AB_303882 | 1:750 dilution |
| Antibody | Anti-CGRP (rabbit polyclonal) | Sigma | Cat# C8198, RRID:AB_259091 | 1:500 dilution |
| Antibody | Anti-POMC (rabbit polyclonal) | Phoenix Pharmaceuticals | Cat# H-029–30 | 1:500 dilution |
| Antibody | Anti-CARTPT (rabbit polyclonal) | Phoenix Pharmaceuticals | Cat# H-003–62 | 1:2000 dilution |
| Antibody | APC–conjugated anti-mouse CD31 (mouse monoclonal) | BioLegend | Cat# 102409 | 1:800 dilution |
| Antibody | APC-anti-mouse CD45 (mouse monoclonal) | BioLegend | Cat# 103111 | 1:800 dilution |
| Antibody | APC-anti-mouse F4/80 (mouse monoclonal) | BioLegend | Cat# 123115, clone BM8 | 1:800 dilution |
| Antibody | PE-Cy7-conjugated anti-mouse CD326 (mouse monoclonal) | Thermo Fisher Scientific | Cat# 25-5791-80, clone G8.8 | 1:400 dilution |
| Antibody | Anti-IgG A488 (goat polyclonal) | Invitrogen | Cat# A11034 | 1:250 dilution |
| Antibody | Anti-IgG A555 (goat polyclonal) | Invitrogen | Cat# A21229 | 1:250 dilution |
| Antibody | Anti-IgG A633 (goat polyclonal) | Invitrogen | Cat# | 1:250 dilution |
| Antibody | Anti-rat IgG (donkey polyclonal) | Jackson ImmunoResearch | Cat# 712-605-153 | 1:250 dilution |

*Continued on next page*

*Continued*

| Reagent type (species) or resource | Designation | Source or reference | Identifiers | Additional information |
|---|---|---|---|---|
| Biological sample (*Homo sapiens,* male and female) | Lung tissue | Stanford University | NA | NA |
| Biological sample (*H. sapiens,* female) | Human pulmonary carcinoid tissue | Stanford University | NA | NA |
| Strain, strain background (*Mus musculus*, male and female) | Wild type: CD-1 and C57BL/6NJ | Charles River | Strain Code 022, Strain Code 027 | NA |
| Strain, strain background (*M. musculus*, male and female) | *Ascl1*$^{CreERT2}$ | *Kim et al., 2011a*. PMID:21483754 | NA | NA |
| Strain, strain background (*M. musculus*, male and female) | Agtr1a-Cre; B6(C3)-*Agtr1a*$^{tm1.1(cre)Ekrs}$/J | The Jackson Laboratory | JAX: 030553 | NA |
| Strain, strain background (*M. musculus*, male and female) | *Agtr1a-2A-Cre; B6.FVB-Agtr1a*$^{em1(cre)Zak}$/J | The Jackson Laboratory | JAX: 031487 | NA |
| Strain, strain background (*M. musculus*, male and female) | *Calb1-2A-dgCre: B6.Cg-Calb1tm1.1(folA/cre)Hze* | The Jackson Laboratory | JAX: 023531 | NA |
| Strain, strain background (*M. musculus*, male and female) | Mouse: *Rosa26*$^{Zsgreen1}$, B6.Cg-Gt(ROSA)26Sor$^{tm6(CAG-ZsGreen1)Hze}$/J | *Madisen et al., 2009* | Stock No: 007906 | NA |
| Commercial assay or kit | RNAscope Hiplex in situ hybridization assay | Advanced Cell Diagnostics (ACD) | Cat#324140 | NA |
| Commercial assay or kit | RNAscope Hiplex assay probe Mm-Agt-T1 | Advanced Cell Diagnostics (ACD) | Cat#426941-T1 | NA |
| Commercial assay or kit | RNAscope Hiplex assay probe Mm-Nmb-T2 | Advanced Cell Diagnostics (ACD) | Cat#459931-T2 | NA |
| Commercial assay or kit | RNAscope Hiplex assay probe Mm-Adcyap1-T3 | Advanced Cell Diagnostics (ACD) | Cat#405911-T3 | NA |
| Commercial assay or kit | RNAscope Hiplex assay probe Mm-Cartpt-T5 | Advanced Cell Diagnostics (ACD) | Cat#432001-T5 | NA |
| Commercial assay or kit | RNAscope Hiplex assay probe Mm-Pomc-T6 | Advanced Cell Diagnostics (ACD) | Cat#314081-T6 | NA |
| Commercial assay or kit | RNAscope Hiplex assay probe Mm-Chga-T9 | Advanced Cell Diagnostics (ACD) | Cat#447851-T9 | NA |
| Commercial assay or kit | RNAscope Hiplex assay probe Mm-Calca-T7 | Advanced Cell Diagnostics (ACD) | Custom probe | NA |
| Commercial assay or kit | RNAscope Hiplex assay probe Mm-Scg5-T10 | Advanced Cell Diagnostics (ACD) | Custom probe | NA |
| Software, algorithm | Integrative Genomics Viewer (IGV v2.4.14) | *Katz et al., 2010*. PMID:21057496 | https://software.broadinstitute.org/software/igv/ | NA |
| Software, algorithm | Seurat v2.3.4 | *Butler et al., 2018*. PMID:29608179 | https://satijalab.org/seurat/ | NA |

## Data and code availability

Further information and requests for resources and reagents should be directed to and will be fulfilled by the lead author Christin Kuo (ckuo@stanford.edu) or lead contact Mark Krasnow (kransow@stanford.edu). All raw and processed data with accompanying metadata from scRNA-seq of mouse and human PNECs will be submitted to the Gene Expression Omnibus (GEO) database https://www.ncbi.nlm.nih.gov/geo. Raw data consists of fastq files corresponding to paired-end reads for each cell, and processed data is in the form of a raw gene counts matrix. All code used to generate data objects and plots is available at github: https://github.com/sdarmanis/Neuroendocrine_scRNA-seq copy archived

at swh:1:rev:c411aa9cce13aba895d521ed979f8a7b47e2d1bd (*Kuo, 2022*) and http://github.com/cskuo/NE_scRNA-seq.

## Experimental model and subject details

### Animals

Mouse lines used were: wild-type mouse strains CD-1 and C57BL/6NJ, aged 2–4 months, tamoxifen-inducible, PNEC-specific knock-in Cre recombinase driver *Ascl1^CreERT2* (*Kim et al., 2011a*) and Cre-dependent fluorescent reporter *Rosa26^LSL-ZsGreen* (*Madisen et al., 2009*), and knock-in Cre recombinase drivers *Agtr1a^Cre* (*de Kloet et al., 2017*), *Agtr1a-2A-Cre* (*Leib et al., 2017*), and *Calb1-2A-dgCre* (*Evans et al., 2017*). Mice were maintained in 12 hr light/dark cycle with food and water provide ad libitum. Genotyping was performed on tail clips with oligonucleotide primers as described (*Kim et al., 2011a*). All animal husbandry, maintenance, and experiments were performed in accordance with Stanford University's IACUC-approved protocols (APLAC 9780, 32092).

### Mouse PNEC labeling, enrichment, and isolation

Mouse PNECs were permanently labeled with ZsGreen by intraperitoneal injection of PN15 (n=4 mice, gender-balanced), PN85 (n=2), and PN115 (n=2) *Ascl1^CreER/+*; *Rosa26^LSL-ZsGreen* mice with 3 mg tamoxifen (MilliporeSigma T5648; prepared at 20 mg/ml in corn oil, stored at –20 °C) to induce Cre-ERT2, and 3–7 days later (PN21, PN90, and PN120) euthanized by $CO_2$ inhalation and cervical dislocation. For the PN21 time point, a second dose of tamoxifen (3 mg) was provided to the dam at E13.5 when PNEC progenitors robustly express *Ascl1* (*Kuo and Krasnow, 2015*). Immediately after euthanasia, lungs (unperfused) were micro-dissected en bloc at room temperature, and the trachea and peripheral regions of each lobe removed with a razor blade. The remaining bronchiolar regions (where PNECs reside) of individual lobes were minced with a razor blade and then digested in DMEM/F12 media containing dispase at 1 U/ml (StemCell Technologies, 07923), type 4 collagenase at 10 U/ml (Worthington Biochemical, CLS-4), and elastase at 3 U/ml (Worthington Biochemical) at 37 °C for 30–60 minutes. Samples were manually triturated with a 1000 µl micropipet tip every 10 minutes during the incubation to generate a uniform cell suspension. An equal volume of cold (4 °C) FACS buffer [phosphate buffered saline (PBS, 137 mM NaCl, 2.7 mM KCl, 2 mM ethylenediaminetetraacetic acid (EDTA), 8 mM $Na_2HPO_4$, 2 mM $KH_2PO_4$, pH 7.4) with 10% fetal bovine serum (Thermo Fisher 10082147)] was added to quench the enzymatic reactions. All subsequent steps were carried out at 4 °C. DnaseI (Stem Cell Technologies, 07900) was added to a final concentration of 5 µg/ml with mixing by tube inversion ten times during the 5-min incubation. Cell suspensions were then filtered sequentially through 100 µm (Corning 431752) and 40 µm mesh filters (Corning 352340) to remove cell aggregates, then centrifuged at 400xg for 5 min. Cell pellets were resuspended in 2 ml of 1 x RBC lysis buffer (BD Biosciences 555899) and incubated for 5 min to deplete red blood cells (RBCs). FACS buffer (8 ml) was added to terminate the reaction, and the suspension centrifuged at 400xg for 5 min, and cell pellets were resuspended in FACS buffer to $10^5$–$10^6$ cells/ml. To deplete endothelial cells and monocytes, 10 µl anti-CD31 MicroBeads (Miltenyi Biotec, 130-097-418) and 10 µl anti-CD45 MicroBeads (Miltenyi Biotec, 130-052-301) were added to 1 ml of cell suspension, incubated for 15 min, and loaded on an LD MACS column (Miltenyi Biotec, 130-042-901) pre-equilibrated with FACS buffer, according to manufacturer's instructions. The column flow-through was centrifuged at 400xg for 5 min, and the cell pellet was resuspended in 0.8–1 ml of FACS buffer to $10^6$–$10^7$ cells/ml.

The cell suspensions were incubated with the following antibodies: allophycocyanin (APC)–conjugated anti-mouse CD31 (BioLegend, 102409, 1:800 dilution), APC-anti-mouse CD45 (BioLegend, 103111, 1:800), APC-anti-mouse F4/80 (BioLegend 123115, clone BM8, 1:800 dilution), phycoerythrin and cyanin 7 (PE-Cy7)-conjugated anti-mouse CD326 (anti-EpCam) (Thermo Fisher Scientific, 25-5791-80, clone G8.8, 1:400 dilution). After 15 min, cells were centrifuged at 300xg x 5 min, the cell pellet was resuspended in 1 ml FACS buffer, and this wash step was repeated twice. The final cell suspension was flow-sorted in a FACSArialI (Becton-Dickinson) using the indicated sorting gates (*Figure 1—figure supplement 1*). Cells were collected in 96-well plates (BioRad, HSP9631) containing 4 µl per well of cell lysis buffer containing 4 U RNase Inhibitor (Takara Bio, 2313 A), 0.05% Triton X100 (Thermo Fisher), 2.5 mM dNTPs (Thermo Scientific) and 2.5 µM oligo-dT30VN (AAGCAGTGGTAT CAACGCAGAGTACT30 VN-3') (IDT) as previously described for Smart-seq2 (*Picelli et al., 2014*), where 'V' represents A,C,G, 'N' represents A,T,C, or G, and the synthesized product contains a mix of

all possible combinations in approximately equal proportions (variance up to 10%). Plates with sorted cells were sealed with microplate sealing film, vortexed 3–5 s, centrifuged at 1000xg for 1 min, and immediately placed on dry ice and stored at –80 °C until complementary DNA (cDNA) generation and sequencing.

## Single-cell mRNA sequencing

RNA from individual sorted cells was reverse transcribed to cDNA amplified, and Illumina sequencing libraries prepared as previously described (*Darmanis et al., 2017*). Briefly, 96-well plates containing single-cell lysates were thawed on ice, heated to 72 °C for 3 min and immediately put back on ice. For cDNA synthesis, 6 µl of reverse transcriptase mix (1 X First-Strand Buffer (Takara Bio, 639538) with 100 U SMARTScribe Reverse Transcriptase (Takara Bio, 639538), 10 U Recombinant RNase Inhibitor (Takara, 2313 A), 8.5 mM DTT (Invitrogen, P2325), 0.4 mM Betaine (Sigma, B0300-5VL), 10 mM $MgCl_2$ (Invitrogen, AM9530G) and 1.6 µM template switching oligonucleotide containing one locked nucleic acid-modified guanosine (+G) at 3'end (5'-AAGCAGTGGTATCAACGCAGAGTACATrGrG+G) (Exiqon)) were added to each well, and the reactions were incubated at 42 °C for 90 min followed by 70 °C for 5 min. For PCR amplification of cDNA, 15 µl of PCR mix (1 x KAPA HiFi HotStart ReadyMix (Kapa Biosystems, KK2602) with 0.16 µM 1SPCR (one step PCR) oligonucleotide (5'-AAGCAGTGGTATCAAC GCAGAGT) (IDT) and 0.56 U of Lambda Exonuclease (New England Biolabs, M0262L)) was added to each well, followed by thermal-cycling at: (i) 37 °C for 30 min, (ii) 95 °C for 3 min, (iii) 21 cycles of 98 °C for 20 s, 67 °C for 15 s and 72 °C for 4 min, and (iv) 72 °C for 5 min. Amplified cDNA was purified using 0.7 x AMPure XP beads (Beckman Coulter, A63880) then analyzed by capillary electrophoresis on a Fragment Analyzer (Advanced Analytical Technologies) and the concentration of cDNA (in fragment size range 500–5000 bp) adjusted and Nextera DNA sequencing libraries prepared as described (*Darmanis et al., 2015*). Libraries from wells on each plate were pooled using a Mosquito liquid handler (SPT Labtech), purified twice using 0.7 x AMPure XP beads, and library pool quality assessed on a Fragment Analyzer. Libraries from 679 single cells were sequenced (75 bp paired-end reads) on a NextSeq 500 (Illumina) using High-output v2 kits (Illumina). Raw sequence reads were demultiplexed using bcl2fastq (v1.8.4, Illumina), and remaining sequences aligned to the mouse reference genome (GRCm38-mm10, UCSC, supplemented with Zsgreen1 sequence) with STAR (v2.5.2b, default parameters except Stranded set to false and Mode set to intersection-nonempty), and the number of reads that mapped to each annotated gene (gene counts) determined with HTSEQ (v0.6.1p1, default parameters except Stranded set to false and Mode set to intersection-nonempty) (*Anders et al., 2015*). Cells with less than 50,000 mapped reads or less than 1000 detected genes were excluded as a quality metric, leaving 534 cell expression profiles for further analysis.

## Analysis of CALCA alternative RNA splicing in PNECs

Sequence reads from both mouse and human PNEC scRNA-seq datasets were aligned to mouse mm10 and human gh38 reference genomes, respectively, using STAR, and the BAM (binary compressed version of sequence alignment map) output visualized by sashimi plots of the Calca (mouse) and CALCA (human) genomic loci using Integrative Genomics Viewer (IGV v2.4.14) (*Katz et al., 2010*; *Robinson et al., 2011*) and the mouse CGRP (RefSeq ID: NM_001289444) and calcitonin (NM_001305616), and human CGRP (NM_001033953) and calcitonin (NM_001741) reference mRNA sequences.

## Validation set of mouse PNEC scRNA-seq profiles

A second set of adult mouse PNECs that were lineage-labeled, purified, and profiled by scRNA-seq as described *Ouadah et al., 2019* was used as validation set. Briefly, PNECs were lineage-labeled by tamoxifen administration to adult (age 2–3 months) $Ascl1^{CreERT2}$;$Rosa26^{LSL-ZsGreen}$ or $CGRP^{CreERT2}$;$Rosa26^{LSL-ZsGreen}$ mice, and whole lungs (excluding trachea) were processed into a single cell suspension. Red blood cells were lysed, and endothelial and immune cells depleted using MACS. Lineage-labeled epithelial cells (ZsGreen+ EpCam+) were sorted by FACS into a single collection tube, and individual cells captured and cDNA generated using an integrated microfluidic platform (Fluidigm C1). cDNA sequencing libraries were prepared in 96 well format and sequenced on a NextSeq 500 (Illumina) device, and obtained sequences were demultiplexed, processed, aligned to individual genes, and quantified to define gene expression levels in each cell. PCA analysis was performed on cell expression patterns using highly variably expressed genes, and identities of cell

clusters with related expression patterns assigned based on enriched expression of canonical lung cell type markers. Of the 100 PNEC expression profiles obtained, the eight with 'transitional' profiles were excluded from our analysis. A threshold of 5 transcripts/million (TPM) was used for determining if a gene was expressed.

## Human PNECs and carcinoid scRNA-seq analysis

The expression profiles of human PNECs characterized here are from our scRNA-seq analysis of cells from histologically normal lung tissue obtained from therapeutic lobectomies and matched blood from three patients with focal lung tumors; these profiles were used to construct our comprehensive molecular cell atlas of the human lung (*Travaglini et al., 2020*). Among the profiled cells, a cluster of 66 PNECs was identified by their selective expression of classical markers CALCA and ASCL1. For our PNEC analysis, we excluded all 11 human PNEC profiles obtained by droplet-based 10 X scRNA-seq, which had less extensive expression profiles than the ones profiled by SS2, the plate-based method used here to profile mouse PNECSs. We also excluded one SS2-profiled cell that was designated a PNEC (cell ID: C7_B002464.gencode.vH29) but was an outlier in the original PNEC cluster *Travaglini et al., 2020*; we found it expressed only one PNEC neuropeptide (DBI) but not any classic PNEC markers (CALCA, CHGA, ASCL1, GRP) or our newly identified PNEC markers (SCGN, PCKS1N, SCG3, SCG5), so it is likely a related but distinct and extremely rare lung cell type. We included in our analysis one SS2-profiled cell that was not originally designated a PNEC (Cell ID: H4.B002460.gencode.vH29), which we found expressed both classic and new PNEC markers (ASCL1, GRP, CHGB, SCGN, SCG2, SCG3, SCG5). In total, our analysis included 55 PNECS, 50 from patient 1 and 5 from patient 3. The SS2 scRNA-seq sequencing reads from these 55 PNECs were re-aligned to the primary assembly of human reference genome GRCH38 (and further analyzed as above), to exclude an alternative contig at the CHGA locus (contig KI270847.1) in the reference genome used in the original analysis (GRCH38.p12) that caused vast undercount of CHGA expression.

The scRNA-seq expression profiles of human carcinoid cells characterized here are from a parallel analysis of one of the tumors in the above study, a typical carcinoid (1.3x0.9 cm) in the left bronchus of the resected left lower lung lobe of patient 3, a 51 year-old female mild adult-onset asthma and recent worsening hypertension. The tumor sample was processed and profiled in parallel with the accompanying normal tissue from this patient, and the 330 cells described here are sorted cells from the epithelial compartment of the tumor sample that were analyzed by SS2 and identified as carcinoid cells by their abundance in the tumor sample and expression of many classic PNEC markers and peptidergic genes, consistent with the underlying clinical and pathological diagnosis. A full description of the carcinoid tumor expression data will be provided elsewhere.

## Peptidergic and sensory genes

The comprehensive list of mouse and human neuropeptide and peptide hormones and their genes ('peptidergic genes'; *Supplementary file 4*) and receptors were compiled from the literature (*Kim et al., 2011b*; *Secher et al., 2016*) and an online database (https://www.neuropeptides.nl/), then verified and updated with newly identified receptors by PubMed literature searches (through July 2020) for each included neuropeptide and peptide hormone.

The comprehensive list of sensory genes (*Supplementary files 6 and 7*) was curated from literature reviews of each sensory modality including mechanosensors (*Clapham, 2003*; *Ranade et al., 2015*), thermosensors (*Caterina et al., 1999*; *McKemy et al., 2002*; *Peier et al., 2002*; *Vandewauw et al., 2018*), acid sensors (*Lin et al., 2004*; *Tominaga et al., 1998*; *Waldmann et al., 1997*), hypoxia sensors (*Chang et al., 2015*; *Kumar and Prabhakar, 2012*), olfactory receptors (*Buck and Axel, 1991*), pheromone receptors (*Dulac and Axel, 1995*), trace amine-associated receptors (*Zucchi et al., 2009*), taste receptors (*Chandrashekar et al., 2006*), and opsins/light sensors (*Blackshaw and Snyder, 1999*; *Haltaufderhyde et al., 2014*; *Terakita, 2005*), and includes all genes with biochemical, genetic, or functional data to support their role as sensors plus related members of the gene family including full families of ion channels (*Yu and Catterall, 2004*). We included all genes previously implicated in PNEC sensory functions including hypoxia-sensing (*Buttigieg et al., 2012*; *Fu et al., 2000*) and mechanosensory (*Lembrechts et al., 2013*; *Lembrechts et al., 2012*) pathways and genes.

## Mouse lung immunohistochemistry and in situ hybridization

For immunohistochemistry, adult wild-type CD-1 or C57BL/6NJ mice as indicated were euthanized as above, and lungs were perfused with room temperature PBS and then inflated with 2% low-melting point agarose (ThermoFisher, UltraPure 16520050). Individual lobes were isolated, fixed at 4 °C for 18–24 hr in 4% paraformaldehyde (PFA) in PBS, cryoprotected in 30% sucrose/PBS solution, transferred to cryomold blocks (22x40 x 20 mm, VWR) and embedded as entire lobes in Optimal Cutting Temperature (O.C.T.) Compound (Tissue Tek), and stored at –80 °C until sectioning. Frozen tissue blocks were sectioned with a cryostat (Leica Biosystems) and the sections (20–50 µm) washed in PBS with 0.1% Tween-20 (PBST) and then incubated with blocking solution (5% goat serum, 0.3% Triton X-100 in PBS) for 1–5 hr, then washed in PBST. Washed sections were incubated with primary antibodies overnight at 4 °C, washed with PBST, and then incubated with secondary antibodies at room temperature for 45 min followed by counterstaining with DAPI at 0.1 µg/ml in PBS to mark nuclei. Primary antibodies were: anti-PC1/Pcsk1 (rabbit, Abcam ab3532, used at 1:750 dilution), anti-CGRP (rabbit, Sigma C8198, 1:500), anti-POMC (rabbit, Phoenix Pharmaceuticals H-029–30, 1:500), anti-Cartpt (rabbit, Phoenix Pharmaceuticals H-003–62, 1:2000). Secondary antibodies were directly conjugated to Alexa-488,–555, or 633 (Invitrogen) or to Alexa 647 (donkey anti-rat, Jackson ImmunoResearch) and used at 1:250 dilution.

For multiplex single molecule FISH (smFISH), wild-type mouse lungs were perfused, inflated, fixed, imbedded in O.C.T. Compound, and stored as above. Cryosections (12 µm) were probed by RNAscope Hiplex12 technology (Advanced Cell Diagnostics, 324140) according to manufacturer's instructions. The proprietary RNAscope probes were: Mm-Agt-T1 (426941-T1), Mm-Nmb-T2 (459931-T2), Mm-Adcyap1-T3 (405911-T3), Mm-Cartpt-T5 (432001-T5), Mm-Pomc-T6 (314081-T6), Mm-Chga-T9 (447851-T9), Mm-Resp18-T11 (493871-T11), Mm-Calca-T7 (custom probe), and Mm-Scg5-T10 (custom probe). Probed sections were imaged by confocal fluorescence microscopy (Zeiss LSM 880, Airyscan mode), and images were aligned using RNAscope HiPlex Registration software and processed with Zen software (Zeiss). To resolve secretory vesicles immunostained for peptides (*Figure 3C and G*), confocal images were acquired in super-resolution mode.

## In vitro imaging of mouse pulmonary sensory neuron response to angiotensin

Pulmonary sensory neurons (PSNs) were prepared from adult (PN120) *Agtr1a*$^{Cre/+}$;*Rosa26*$^{LSL-tdTom/+}$ mice that selectively label the two types of PSNs that innervate NEBs (Liu et al, unpublished data). Three to five days prior to PSN isolation, 50 µl (1 mg/mL) of a fluorescent wheat germ agglutinin (WGA-647, Thermo Fisher, W32466) was instilled into the trachea to retrograde label PSNs. After 3–5 days to allow for WGA uptake by PSNs and retrograde transport to their cell bodies in the tenth (vagus) cranial nerve ganglia, mice were euthanized as above and the vagal ganglia were dissected and immediately placed in cold-buffered Hanks Balanced Salt Solution without calcium or magnesium (HBSS, ThermoFisher, 14190144). Ganglia were digested with 60 U papain (Worthington Biochemical, LS003126) in 1 ml HBSS (with 10 mM HEPES pH 7.4, 0.5 mM EDTA, and 0.4 mg/mL L-cysteine) for 10 min at 37 °C. The papain solution was then replaced with 3 ml of a second enzymatic digestion solution (1.5 mg/ml collagenase IV (Worthington, LS004186) and 1 mg/ml dispase (Worthington, LS02109) in HBSS with 10 mM HEPES) and the incubation continued at 37 °C for 30 min, with tube inversion 5 times every 10 min. The sample was centrifuged for 4 min at 400 g, and the pelleted cells were resuspended in 1 ml L-15 medium (Gibco 11415) with 10 mM HEPES (pH 7.4) by three sequential rounds of manual trituration using custom pulled glass micropipettes (Sutter Instrument Company, Model P-87) of successively finer tip diameter (to final range 0.1–0.12 mm). Tips were pre-coated with complete L-15 medium with 10% FBS (Gemini Bio Products, 100–50, diluted in 10 mM HEPES, pH 7.4) to limit neuronal loss. The cell suspension was gently layered on 5 ml of 20% Percoll (Sigma, P4937) in L-15 medium (Gibco, 11415) and centrifuged for 9 min at 400xg to separate dissociated neurons from lower density connective tissue and smaller cells. The cell pellet was resuspended in 2 ml of L15 medium with 10 mM HEPES (pH 7.4), giving a typical yield of ~1000 cells. Cells were transported to the imaging facility at room temperature and centrifuged for 3 min at 750xg. Cells were re-suspended in 100 µl of warm $CO_2$-equilbrated DMEM/F12 medium (Gibco, 10565018), and 30–40 µl of the cell suspension were plated in the center of a laminin-coated inset of a poly-lysine-coated 12 mm circular Nunc glass bottom culture dish (Thermo Fisher, 150680), prepared as described below. Cells were

incubated at 37 °C for 60–90 min to initiate cell adherence to the inset; although only some neurons adhere during this period, incubations beyond 2 hr caused evaporation of the small volume of medium and decreased neuron viability. DMEM/F12 medium (500 µl) was added to each well, and the cultures were incubated at 37 °C (with 5% $CO_2$) for 12–16 hr to increase cell adherence and equilibrate cells to the culture environment prior to functional imaging. In healthy preparations, typically ~100–150 neurons adhered to the inset and ~10% formed extended projections. (We found that coating of the culture insets with fresh reagents as follows was critical for cell adherence and viability: Poly-lysine coating of clean Nunc glass bottom dishes was done by incubating the dish in a solution of 50 µg/ml poly-D-lysine [Millipore, A-003-E] in HBSS at 37 °C overnight, then washing the dish with HBSS three times and removing residual solution by aspiration; laminin coating of the insets was done by covering insets with a solution of 20 µg/ml laminin [Sigma L2020] in HBSS at 4 °C for at least 45 min [typically 3–4 hr], then carefully removing the laminin solution by aspiration, washing the inset three times with HBSS at 4 °C, and leaving the inset covered in HBSS until cell plating).

For functional imaging, cells were loaded with fluorescent calcium indicator Fluo-4 by incubating cells with 10uµm Fluo-4 (Invitrogen) in HBSS buffered with 10mM HEPES (pH 7.4) for 15–20min. Nunc glass bottom dishes with insets containing cultured neurons were placed on a perfusion chamber platform (Warner Instruments, RC-37W) on a Zeiss 880 LSM confocal microscope stage housed within an incubation chamber adjusted to 37°C and 5% $CO_2$; platform perfusion was by gravity-dependent flow controlled with a stopcock. Cells were continuously perfused with HBSS buffered with 10mM HEPES (pH 7.4), and fields containing retrograde-labeled pulmonary sensory neurons were identified by WGA-647 fluorescence. Calcium imaging data (Fluo-4 flourescence) were collected every second with 488nm wavelength excitation and 500–550nm emission. After baseline recording for 60s, 500nM Angiotensin II peptide (Sigma-Aldrich, A9525) in HBSS with 10mM HEPES (pH 7.4) was perfused for 60s, followed by a wash and re-equilibration with HBSS buffered with 10mM HEPES (pH 7.4) for 3min, and finally a 15-s infusion of 50mM KCl in HBSS buffered with 10mM HEPES (pH 7.4) to assess cell excitability/viability. Following the infusions, the dish was removed from the perfusion chamber and cells immediately fixed with 4% PFA at 4°C for 30–60min for subsequent immunohistochemistry as above to confirm identity of the monitored neurons. For analysis of the time-lapse recordings, Image J software (v2.0) was used to define cell boundaries and determine the mean fluorescence intensity value for each cell in the imaging field at each time point; the obtained fluorescence values were normalized to the average baseline fluorescence value (prior to Angiotensin II exposure) for the cell.

## Quantification and statistical analysis
### Computational clustering and identification of mouse PNEC scRNA-seq profiles

Counts for each gene were normalized across cells, scaled per million and converted to logarithmic scale. Dimensionality reduction was used to compare and cluster the obtained cell expression profiles using Seurat v2.3.4 (*Butler et al., 2018*). First, genes with highly variable expression across the sample population were identified ('FindVariableGenes'), selecting genes with >1 standard deviation dispersion in mean expression values. Second, the dimensionality of the expression matrix data for the highly variable genes was reduced using principal component analysis (PCA), and the significant principal components (PCs) that captured the majority of variation in the dataset were selected by their standard deviations (PCElbowPlot function) and by examining the top gene loadings in each component as heatmaps. We selected the first 15 PCs, and the 5 genes with the highest PC scores along each PC were inspected for biological relevance and for canonical markers of known lung cell types. Then, the relatedness of cell expression profiles was visualized in two-dimensional tSNE plots ('RunTSNE for R' with perplexity = 30). Third, genes enriched in each cluster of cells with similar expression profiles were identified using Wilcoxon rank sum test with multiple testing correction ('FindAllMarkers'). Cell doublets were identified and removed using Scrublet (*Wolock et al., 2019*).

The identities of the PNEC cell cluster and the nine other obtained cell clusters were assigned based on enriched expression of canonical lung cell type markers: PNECs (*Calca, Ascl1, Chga*), multi-ciliated cells (*Foxj1, Ccdc153, Cdhr3*), basal cells (*Krt5, Trp63, Krt15*), AT1 cells (*Ager, Rtkn2*), AT2 cells (*Sftpc, Sftpb*), club cells (*Scgb3a2, Scgb1a1*), endothelial cells (*Pecam, Tie1*), stromal cell populations 1 and 2 (*Col1a1, Col1a2*), and glial cells (*Gfap, S100b, Plp1*). The expression profiles of the 176 obtained high quality PNECs were used for all analyses. To generate the marker list shown in ***Supplementary***

*file 1*, we merged the 534 cells with the mouse lung cell atlas and performed another marker analysis to identify top differentially expressed genes in NE cells compared to non-NE epithelial cells and this analysis of top markers is shown in *Figure 1*.

## Analysis of peptidergic gene expression in human carcinoids profiled by bulk RNA-seq

Bulk RNA-seq datasets of 239 lung neuroendocrine neoplasms (*Alcala et al., 2019*; *Fernandez-Cuesta et al., 2014*; *Laddha et al., 2019*) were obtained from https://nextjournal.com/rarecancersgenomics/a-molecular-map-of-lung-neuroendocrine-neoplasms/ (*Gabriel et al., 2020*), and the histopathologic classification in the attributes metadata file was used to identify the 111 typical and atypical lung carcinoids whose expression profiles of peptidergic genes and PNEC markers were analyzed here. Log-transformed gene read counts were normalized by quantile normalization (*Dillies et al., 2013*), and the obtained values were represented as heatmaps (*Figure 6—figure supplement 6*). Code available at http://github.com/cskuo/NE_scRNA-seq.

## Rarefaction analysis to estimate saturation of PNEC neuropeptide diversity

We modeled each PNEC peptidergic expression profile as an incidence sampling of all total possible peptidergic genes expressed by the PNEC population, and estimated the peptidergic diversity using rarefaction and extrapolation analysis, a technique used in ecology to assess species richness (*Chao et al., 2014*). In this analogy, each 'species' is a peptidergic gene, and a given PNEC that may express any number of distinct peptidergic genes is analogous to a sampling of the total assemblage of species (peptidergic genes). Using the iNEXT package to estimate species richness (*Hsieh et al., 2016*), we constructed an integrated curve to smoothly link rarefaction (interpolation) and prediction (extrapolation), and the associated 95% confidence intervals, by bootstrapping (N=200). Only the incidence (presence or absence), and not the abundance, of each peptidergic gene RNA was used to estimate the underlying neuropeptide accumulation curve (*Chao et al., 2014*; *García-Ortega and Martínez, 2015*).

## Acknowledgements

We thank Kyle Travaglini and Ahmad Nabhan (processing of human carcinoid tumor), Y Ouadah (mouse Fluidigm scRNA-seq data), and all members of Krasnow lab and Jeffrey Wine (discussions and comments on manuscript); Norma Neff and members of the Quake laboratory (single cell expertise and resources); Rong Lu and Stanford Quantitative Sciences Unit (normalization of previously published bulk carcinoid data); Cathy Crumpton, Brandon Carter, and Stanford FACS facility (flow sorting); Joseph Shrager and Jalen Benson (IRB protocol and tissue procurement); the patient in this study; and Maria Peterson (figure preparation), the Pediatric Pulmonary Division, and Department of Pediatrics for space and resources. This work was supported by grants from the NIH/NHLBI (K08HL129081, CSK), Doris Duke Charitable Foundation (2018105, CSK), Howard Hughes Medical Institute (MAK), Ludwig Foundation (MAK), NIH/NCI 5U01CA231851 (MAK, CSK) and the Chan Zuckerberg Initiative (MAK). MAK is an investigator of the Howard Hughes Medical Institute.

## Additional information

### Funding

| Funder | Grant reference number | Author |
| --- | --- | --- |
| National Institutes of Health | K08HL129081 | Christin S Kuo |
| Doris Duke Charitable Foundation | 2018105 | Christin S Kuo |
| Howard Hughes Medical Institute | | Mark A Krasnow |

| Funder | Grant reference number | Author |
| --- | --- | --- |
| National Cancer Institute | U01CA231851 | Christin S Kuo |
| Chan Zuckerberg Initiative | | Mark A Krasnow |

The funders had no role in study design, data collection and interpretation, or the decision to submit the work for publication.

## Author contributions

Christin S Kuo, Conceptualization, Resources, Data curation, Software, Formal analysis, Supervision, Funding acquisition, Investigation, Visualization, Methodology, Writing – original draft, Project administration, Writing – review and editing, Conceived the project; Spyros Darmanis, Methodology, Writing – review and editing, Sorted cells and prepared sequencing libraries, Processed and aligned sequencing data; Alex Diaz de Arce, Investigation, Methodology, Writing – review and editing, Designed and performed pulmonary sensory neuron functional imaging experiments; Yin Liu, Formal analysis, Investigation, Methodology, Writing – review and editing, Provided scRNA-seq receptor gene expression profiles and analysis of pulmonary sensory neurons; Nicole Almanzar, Investigation, Methodology, Writing – review and editing; Timothy Ting-Hsuan Wu, Investigation, Methodology, Writing – review and editing, Performed rarefaction analysis; Stephen R Quake, Resources, Supervision, Investigation, Writing – review and editing; Mark A Krasnow, Conceptualization, Resources, Formal analysis, Supervision, Funding acquisition, Visualization, Writing – original draft, Project administration, Writing – review and editing

## Author ORCIDs

Christin S Kuo http://orcid.org/0000-0001-9227-5578
Timothy Ting-Hsuan Wu http://orcid.org/0000-0002-6869-362X
Stephen R Quake http://orcid.org/0000-0002-1613-0809
Mark A Krasnow http://orcid.org/0000-0002-1976-5471

## Ethics

Human subjects: Patient tissues were obtained under protocol and de-identified. Research approved by Stanford University's Human Subjects Research Compliance Office (IRB 15166), and informed consent was obtained from each patient prior to surgery.

This study was performed in strict accordance with the recommendations in the Guide for the Care and Use of Laboratory Animals of the National Institutes of Health. All animal husbandry, maintenance, and experiments were performed in accordance with Stanford University's IACUC-approved protocols (APLAC 32092, APLAC 9780).

## Decision letter and Author response

Decision letter https://doi.org/10.7554/eLife.78216.sa1
Author response https://doi.org/10.7554/eLife.78216.sa2

# Additional files

## Supplementary files

• Supplementary file 1. Mouse airway neuroendocrine cell markers.

• Supplementary file 2. Extant PNEC markers.

• Supplementary file 3. Expression of neurotransmitter biosynthetic, vesicular loading and reuptake genes in mouse PNECs.

• Supplementary file 4. Summary of mouse and human peptidergic gene expression in PNECs.

• Supplementary file 5. PNEC peptidergic genes, receptors, and classic functions.

• Supplementary file 6. Sensory genes (mouse). (a) Mechanical. (b) Thermal. (c) Mrgprs. (d) Acid. (e) Oxygen. (f) CO2. (g) TRP family. (h) Olfactory. (i) Pheromone. (j) Taste. (k) K2P channels. (l) Kv channels.

• Supplementary file 7. Sensory genes (human). (a) Mechanical. (b) Thermal. (c) Mrgprs. (d) Acid. (e) Oxygen. (f) CO2. (g) TRP family. (h) Olfactory. (i) Pheromone. (j) Taste. (k) K2P channels. (l) Kv channels.

• Supplementary file 8. Expression of neurotransmitter biosynthetic, vesicular loading and reuptake genes in human PNECs.

• MDAR checklist

## Data availability

Sequencing data have been deposited in GEO under accession code GSE191178.

The following dataset was generated:

| Author(s) | Year | Dataset title | Dataset URL | Database and Identifier |
|---|---|---|---|---|
| Kuo CS, Darmanis S, Quake SR, Krasnow MA | 2022 | Neuroendocrinology of the lung revealed by single cell RNA sequencing | https://www.ncbi.nlm.nih.gov/geo/query/acc.cgi?acc=GSE191178 | NCBI Gene Expression Omnibus, GSE191178 |

The following previously published dataset was used:

| Author(s) | Year | Dataset title | Dataset URL | Database and Identifier |
|---|---|---|---|---|
| Gabriel AAG, Mathian E, Mangiante L, Voegele C, Cahais V, Ghantous A, McKay JD, Alcala N, Fernandez-Cuesta L, Foll M | 2020 | Supporting data for "A molecular map of lung neuroendocrine neoplasms | http://gigadb.org/dataset/100781 | GigaDB, 10.5524/100781 |

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
