## [Editor Report]

The authors present a comprehensive profile of signals and sensors expressed in mouse and human PNECs by single-cell RNA sequencing. Analyses revealed a myriad combination of neuropeptide, neurotransmitter, receptor and channel genes in PNECs. The authors also surveyed cognate receptors expressed in epithelial cells, endothelial cells, stromal cells, immune cells, and pulmonary sensory neurons, identifying potential local targets for the PNECs signals. They showed one new signal, angiotensin II, directly activates a subpopulation of innervating pulmonary sensory neurons that project to the brain. The scRNA-seq profile of a lung carcinoid tumor suggests that selected PNECs are susceptible to carcinoid transformation and together, these data indicate that PNECs serve as sentinels to perceive multiple airway stimuli and express a variety of signals that either act locally, through pulmonary sensory neurons, or potentially through circulation to regulate homeostasis.

---

## [Decision Letter]

**Decision letter after peer review:**

Thank you for submitting your article "Neuroendocrinology of the lung revealed by single-cell RNA sequencing" for consideration by *eLife*. Your article has been reviewed by 3 peer reviewers, and the evaluation has been overseen by Paul Noble as the Senior and Reviewing Editor. The following individual involved in the review of your submission has agreed to reveal their identity: Michael Shen (Reviewer #2).

Essential revisions:

All three reviewers acknowledged the merit of the study from this outstanding group, but suggest additional clarification and function studies be performed. Please review the public reviewer comments when revising the manuscript and incorporate the salient points. This study catalogs the transcriptomic diversity of PNECs. Agt action on PSN neurons was included as a functional test. Additional functional experiments will enhance the impact of the study and warrant publication in this journal:

1. A rigorous definition of neuroendocrine cells should be stated clearly and applied consistently across all datasets. If cells are likely to be neuroendocrine but do not meet the complete definition (e.g., ZsGreen+ EpCAM- cells), then it might be helpful to indicate them in the figure(s) and explain how they do or do not differ in their properties from the rigorously-defined subset of PNECs.

2. The presentation of sensitivity and specificity of PNEC gene expression would benefit from a different graphical format. For example, some type of two-dimensional plots might illustrate these points about peptidergic gene expression more clearly.

3. Greater clarity on the number of samples examined and the addition of statistical analyses would be very helpful to include.

4. Pointing out which genes are expressed in PNECs but not in TNECs will be informative.

5. The statement that most peptidergic genes are "expressed in rare PNEC subpopulations" needs to be based on smFISH data, and not just scRNA-seq data given the drop-off effects. The lack of reads in the expected potassium channels also warrants further investigation by smFISH.

6. Neuropeptides are typically enriched in vesicles on the basal side of PNECs, and this is reflected in most of the staining results. However, in Figure 3C, this does not appear to be the case, which warrants re-investigation.

7. In Figure 4F, neuronal traces in the lower panel have 3 different colors (grey, dark grey, and black), what are the numbers of cells in each group? And what may explain why Agtr1a+ PSN7 neurons were not activated after Agt II application? Are these neurons activated at higher concentrations?

8. Agt is expressed by other cell types in the lung. Do Agtr1a+ neurons only target PNECs? What is the evidence that PNECs are the source of Agt that activates Agtra1 neurons?

9. Are there expression pattern differences between solitary and clustered PNECs in either mouse or human?

10. Is there any information in the single cell data that would explain the cause for rare PNEC types being amplified in the carcinoid?

11. Description of signature and staining localization of glial cells will be informative.

*Reviewer #1 (Recommendations for the authors):*

The authors should consider revising the title to give it encompass the overall conclusion and the context.

In the Abstract section:

It would be better to write the expanded form of the abbreviation at the first place throughout the text, for example, CGRP and GABA in the second line of the abstract.

The authors' statement "release presumably regulated by the distinct, multimodal combinations of sensors expressed" needs to have specificity with examples. What sort of sensors?

The authors' statement "Expression of cognate receptors" should include specific examples.

In the Introduction section:

Line 5: related functions should include a few specific examples.

Line 8-9: "..several neuropeptides including CGRP, which can signal locally to lung goblet (GABA) and immune cells (CGRP).." the abbreviation CGRP should be expanded at its first appearance. Also, the sentence should be re-phrased to explain the context of GABA and CGRP shown in brackets.

Line 12: In the sentence, "they can be transformed by loss of tumor suppressor genes", please mention a few examples of the tumor suppressor genes.

In the Results section:

Section 2: The sentence "PNECs express dozens of neuropeptide and peptide hormone genes" does not seem to be scientifically sound. Authors should consider revising it to infer a concise sub-heading for this part.

Section 2: In the sentence "suggesting they can signal through serotonin imported from other sources.", the other source should be defined with precise citation, or is it just the authors' assumption?

Section 2: Along with the names of genes, authors should consider citing the references vouching for neurotransmitters' type-specific genes in the sentence "PNECs may also use other neurotransmitters because rare cells expressed key dopaminergic (Th, Ddc), glutamatergic (Slc17a6), cholinergic (Slc18a3/VAChT), or histaminergic (Hdc) genes (Table S3).".

(A general comment/suggestion due to redundant appearance) The appearance of the phrase "dozens of genes" should be avoided throughout the text. Alternatively, authors can consider being specific about the numbers in their dataset.

Authors are recommended to cite specific references for 17 PNEC peptidergic genes; based on what these genes have been selected?

In the sentence "over an order of magnitude more than previously known", authors are recommended to include the previously known percentage or the number of peptidergic genes precisely.

The Results sections 2 and 3 are overlapping. Authors should consider distinguishing the context.

The authors should consider either combining them or segregating the talk about the peptidergic genes from the part of neuropeptide and peptide hormone genes.

In section 4, the sentence "PNECs are diverse" should have specificity about the type. Depending on what, the PNECs are diverse. To me, the results seem to be direct towards the PNEC diversity in the context of their functions. Authors should consider editing the sub-heading to depict a more specific context.

In section 5, again there are too many overlaps and redundancy of the information with sections 2 and 3. Authors should consider to re-arrange the result parts to give a concise and meaningful context.

In section 6, the sentence "The human lung cell expression patterns of receptors for the 32 PNEC peptidergic signals with known receptors are shown in Figure S10A, identifying potential direct targets in lung." What are the direct targets the authors are referring to?

In section 6, the sentence "The predicted targets of the conserved PNEC signals were also largely conserved, for example, broad stromal and vascular targeting by inhibin, immune cell targeting by CGRP and VGF, and pericyte targeting by angiotensin.", the claim should also be supported with references.

In section 7, the sentence "However, they did not recapitulate the full spectrum of PNEC diversity in their sensory and signaling gene profiles." Authors should consider explaining it briefly with specific citations.

*Reviewer #2 (Recommendations for the authors):*

The manuscript would be considerably strengthened by the following:

1. A rigorous definition of neuroendocrine cells should be stated clearly and applied consistently across all datasets. If cells are likely to be neuroendocrine but do not meet the complete definition (e.g., ZsGreen+ EpCAM- cells), then it might be helpful to indicate them in the figure(s) and explain how they do or do not differ in their properties from the rigorously-defined subset of PNECs.

2. The presentation of sensitivity and specificity of PNEC gene expression would benefit from a different graphical format. For example, some type of two-dimensional plots might illustrate these points about peptidergic gene expression more clearly.

3. Greater clarity on the number of samples examined and the addition of statistical analyses would be very helpful to include.

*Reviewer #3 (Recommendations for the authors):*

This study catalogs the transcriptomic diversity of PNECs. Agt action on PSN neurons was included as a functional test. Additional functional experiments will enhance the impact of the study and warrant publication in this journal.

1. Pointing out which genes are expressed in PNECs but not in TNECs will be informative.

2. The statement that most peptidergic genes are "expressed in rare PNEC subpopulations" needs to be based on smFISH data, and not just scRNA-seq data given the drop-off effects. The lack of reads in the expected potassium channels also warrants further investigation by smFISH.

3. Neuropeptides are typically enriched in vesicles on the basal side of PNECs, and this is reflected in most of the staining results. However, in Figure 3C, this does not appear to be the case, which warrants re-investigation.

4. In Figure 4F, neuronal traces in the lower panel have 3 different colors (grey, dark grey, and black), what are the numbers of cells in each group? And what may explain why Agtr1a+ PSN7 neurons were not activated after Agt II application? Are these neurons activated at higher concentrations?

5. Agt is expressed by other cell types in the lung. Do Agtr1a+ neurons only target PNECs? What is the evidence that PNECs are the source of Agt that activates Agtra1 neurons?

6. Are there expression pattern differences between solitary and clustered PNECs in either mouse or human?

7. Is there any information in the single cell data that would explain the cause for rare PNEC types being amplified in the carcinoid?

8. Description of signature and staining localization of glial cells will be informative.

---

## [Author Response]

Essential revisions:All three reviewers acknowledged the merit of the study from this outstanding group, but suggest additional clarification and function studies be performed. Please review the public reviewer comments when revising the manuscript and incorporate the salient points. This study catalogs the transcriptomic diversity of PNECs. Agt action on PSN neurons was included as a functional test. Additional functional experiments will enhance the impact of the study and warrant publication in this journal:1. A rigorous definition of neuroendocrine cells should be stated clearly and applied consistently across all datasets. If cells are likely to be neuroendocrine but do not meet the complete definition (e.g., ZsGreen+ EpCAM- cells), then it might be helpful to indicate them in the figure(s) and explain how they do or do not differ in their properties from the rigorously-defined subset of PNECs.

We define pulmonary neuroendocrine cells (PNECs) throughout the paper by their transcriptomic clustering and signatures, which includes the dozens of newly identified PNEC markers as well as the few extant marker genes available before this study (listed in Supplementary file 2). The confusion here arises from the two previously known markers (*Ascl1* lineage marker ZsGreen, EpCAM) we used for flow sorting to enrich for these rare cells for transcriptomic profiling (Figure 1—figure supplement 1). Although most of the cells with PNEC transcriptomic profiles were from the ZsGreen^hi^ EpCAM^hi^ sorted population (as expected), some were from the ZsGreen^hi^ EpCAM^lo^ sorted population. The latter resulted from the EpCAM gating threshold we used during flow sorting, which excluded some PNECs with intermediate levels of surface EpCAM from the EpCAM^hi^ population. Indeed, nearly all PNECs (> 95%) expressed *Epcam* by scRNAseq, and there was no difference in *Epcam* transcript levels or transcriptomic clustering of PNECs that were obtained from the ZsGreen^hi^ EpCAM^hi^ vs. ZsGreen^hi^ EpCAM^lo^ sorted populations, as we now show in the two panels (C', C'') added to Figure 1—figure supplement 1C. This point is now clarified in the legend to Figure 1—figure supplement 1C, and it nicely demonstrates that transcriptomic profiling is a more robust method of identifying PNECs than flow sorting based on two classical markers.

2. The presentation of sensitivity and specificity of PNEC gene expression would benefit from a different graphical format. For example, some type of two-dimensional plots might illustrate these points about peptidergic gene expression more clearly.

As suggested, we added a two-dimensional plot showing sensitivity and specificity of PNEC markers (Figure 1—figure supplement 1E). Because different marker features are important for different applications, to aid in selection of optimal markers we have provided several different graphical formats (Figures 1B,C, Figure 1—figure supplement 1E) and a table (Supplementary file 1) for comparing markers. This point is discussed further below in the response to point 3 of Reviewer 2.

3. Greater clarity on the number of samples examined and the addition of statistical analyses would be very helpful to include.

As detailed below in the response to points 5 and 6 of Reviewer 2, the number of samples analyzed for the smFISH and immunostaining analyses have been added (Figures 2B, 3F, 3G), as has the requested statistical analysis of the data in Figures 1E, 1G.

4. Pointing out which genes are expressed in PNECs but not in TNECs will be informative.

In the original manuscript we noted tracheal neuroendocrine cells (TNEC) markers (from the mouse tracheal cell transcriptomic atlas by Montoro et al., Nature 2018) that were not expressed in our profiled mouse PNECs. The converse (PNEC markers not detected in TNECs) are also of interest but more problematic to identify from the available data because the droplet-based scRNA-seq method used for profiling TNECs (in Montoro et al) is subject to greater technical dropout (no detection of mRNA from an expressed gene) than the plate-based profiling method we used for PNECs. Nevertheless, as detailed below in the response to point 1 of Reviewer 3, using stringent criteria that take dropout into account we were able to identify several genes (*Igf2, Piezo2, Ptprn, Meg3*) that are likely PNEC-selective markers. We have added this analysis to the paper (p. 5, Figure 1—figure supplement 1F, Supplementary file 1).

5. The statement that most peptidergic genes are "expressed in rare PNEC subpopulations" needs to be based on smFISH data, and not just scRNA-seq data given the drop-off effects. The lack of reads in the expected potassium channels also warrants further investigation by smFISH.

We agree that technical drop-out is a consideration here as for any scRNA-seq dataset, and we addressed this issue in two ways. First, we analyzed a second PNEC scRNAseq dataset (Ouadah et al., 2019) in which PNECs were labeled, isolated, and processed by different methods (including microfluidic capture) but deeply sequenced like ours, and obtained similar results across the full set of peptidergic genes (Figure 1E, Supplementary file 4). Second, we used state-of-the art multiplex single molecule fluorescence in situ hybridization (smFISH) to analyze the expression in PNECs of eight peptidergic genes, including ones expressed in most or all PNECs (2 genes), some PNECs (2 genes), or only rare PNECs (4 genes) in the scRNA-seq analysis. This confirmed the peptidergic gene abundance results from the scRNA-seq analysis (Figure 1G), including the finding that many of the newly identified PNEC peptidergic genes are expressed only in rare subpopulations of PNECs (and that the latter result is not an artifact of technical dropout in scRNA-seq). This experiment (Figure 1G) also showed that smFISH is more sensitive than our scRNA-seq analysis in detecting gene expression, but only if quantification includes cells with only a few (1 – 4) expression puncta (close to the background level). We agree that, ideally, expression of each of the 37 other newly identified PNEC peptidergic genes as well as PNEC potassium channel genes (and the dozens of other genes we identified of biological interest in our PNEC scRNA-seq analysis) should be validated by smFISH, but at current costs (~$1000/gene) it would be prohibitively expensive and take months to analyze and quantify because PNECs are extremely rare (and minor subpopulations proportionately rarer) so laborious to find and quantify in serial lung sections. We feel that our smFISH experiments validating the expression levels and quantifying the abundance of expressing cells for a representative subset of identified PNEC genes (eight peptidergic genes spanning the full range of abundance in PNECs and one new PNEC marker *Resp18*) is a reasonable number for this stage (and cost) of the technology.

Potassium channel genes are not the focus of this paper so were not prioritized for smFISH confirmation or follow up studies. In one of the Results sections ("PNECs are diverse, multimodal sensors"), we noted in the paragraph on PNEC hypoxia sensing that two potassium channel genes (*Kcnc3, Kcnd4*), which prior studies suggested might encode a PNEC hypoxia-sensitive potassium current (Youngson et al. 1993, Fu et al., 2002; reviewed in Garg et al., 2019), were expressed at a low level (*Kcnc3*) or in only rare (*Kcnd4*) mouse PNECs, whereas many other potassium channel genes were expressed more robustly and selectively in the PNECs (Figure 5—figure supplement 1BC <Figure S6B>). Prior studies (Wang et al. 1996; Cutz et al., 2009) assessed expression in PNECs (by conventional in situ hybridization or RT-PCR) of a few potassium channel genes (including *Kcnc3*, *Kcnd4*) and only in fetal or infant lungs from rabbit and human so are not directly comparable to our adult mouse PNEC data (Figure 5—figure supplement 1BC). Identifying the channel(s) that mediate the proposed PNEC hypoxia-sensitive potassium current will require substantial follow up on the expression and function of each PNEC potassium channel gene. To help prioritize genes for such studies, we added a similar analysis of potassium channel gene expression in adult human PNECs (Figure 6—figure supplement 5B). Although beyond the scope of this paper, future studies of the expressed genes should identify the physiologically relevant PNEC channels and any stage- and species-specific expression and function.

6. Neuropeptides are typically enriched in vesicles on the basal side of PNECs, and this is reflected in most of the staining results. However, in Figure 3C, this does not appear to be the case, which warrants re-investigation.

Figure 3C is a super resolution confocal optical section through an immunostained PNEC cluster that was selected to show that CGRP and calcitonin are co-expressed in individual PNECs but localize to distinct secretory vesicles. This was easiest to visualize in the apical region of these PNECs, where individual puncta (vesicles) were most readily resolved. The original image in Figure 3C was an oblique section of a PNEC with an extreme morphology (substantial apical domain with a slender projection to the basement membrane). We have now replaced it with a section that captures the full apicobasal extent of two PNECs with more typical columnar epithelial structures, which shows both the sub-cellular distribution of the individual peptides as well as their largely distinct vesicles.

7. In Figure 4F, neuronal traces in the lower panel have 3 different colors (grey, dark grey, and black), what are the numbers of cells in each group? And what may explain why Agtr1a+ PSN7 neurons were not activated after Agt II application? Are these neurons activated at higher concentrations?

To clarify Figure 4F, we now show the calcium-imaging traces of the three overlaid examples of PSN4 neurons (red traces), other PSNs (black traces), and non-PSNs (grey traces) in separate parts of the panel (top, middle, and bottom of panel F, respectively); the individual traces are provided in Figure 4—figure supplement 2. The results for all 189 viable PSNs analyzed across the 11 experiments are summarized in the source data file. The only other *Agtra1a+* pulmonary sensory neuron type is PSN7, which express lower levels of *Agtra1a* receptor gene than the PSN4 neurons; this could explain the selective activation of PSN4 neurons by Agt II, as we now note in the text (p. 12). However, because the results with PSN7 neurons are negative (no response to Agt II, but viable and responsive to KCl control, see Figure 4—figure supplement 2) including a trial at twice the standard Agt II concentration, there could be other explanations (e.g., the isolated PSN7 neurons analyzed were not fully healthy), so we consider the PSN7 result provisional and note this caveat in the text (p. 12).

8. Agt is expressed by other cell types in the lung. Do Agtr1a+ neurons only target PNECs? What is the evidence that PNECs are the source of Agt that activates Agtra1 neurons?

Across the entire mouse lung cell atlas, the only cell types with prominent expression of angiotensinogen (*Agt*) are PNECs and myofibroblasts (Figure 4—figure supplement 2). The only known lung cell target of *Agtr1a*-expressing pulmonary sensory neurons (PSN4, PSN7) are PNECs (p. 11), which directly contact *Agtr1a*-expressing pulmonary sensory neuron termini (Figure 4D), supporting our model that they provide the Agt ligand to the receptor-expressing PSN4. There are also occasional alveolar projections from PSN4 that do not appear to target a specific cell type (Diaz de Arce et al., in preparation), but it is possible that ligand secretion from alveolar myofibroblasts could also activate PSN4 through these alveolar termini, as we now note in the legend to Figure 4—figure supplement 2.

9. Are there expression pattern differences between solitary and clustered PNECs in either mouse or human?

That is an interesting question but we have not yet identified such differences, although we are actively searching. Nothing obvious emerged from the initial analysis of PNEC scRNA-seq profiles, which likely include profiles of both clustered and solitary PNECs since our PNEC isolation strategy labels and should recover both. Identification of such expression differences will require careful comparison of the expression patterns of PNECs in situ. We have begun analyzing in this way some of the PNEC genes expressed in PNEC subsets, but so far have not found any that distinguish solitary from clustered PNECs. Such distinguishing markers must be rare.

10. Is there any information in the single cell data that would explain the cause for rare PNEC types being amplified in the carcinoid?

No, there is nothing obvious. Indeed, that prompted our speculation that sporadic oncogenic mutations in various PNECs can transform them and cause their amplification into carcinoid tumors, with the amplified cells retaining a memory (albeit imperfect) of the transformed cell's peptidergic expression profile (p. 20). This would explain the diverse endocrine effects of pulmonary carcinoid tumors found in different patients, a consequence that follows from this model and the large and diverse peptidergic profiles we uncovered for PNECs (Figures 6D, Figure 6—figure supplement 6)*.* However, because some peptidergic genes that are expressed in only minor subpopulations of normal PNECs were detected in many carcinoids, whereas a nearly ubiquitous PNEC peptidergic gene (*CALCA*) was absent from most, it appears that not all normal PNECs are equally susceptible to carcinoid transformation (p. 20).

11. Description of signature and staining localization of glial cells will be informative.

We agree that profiles and localization of pulmonary glial cells are of general interest, however since they are unrelated to the topic of this manuscript (neuroendocrine cells) that information forms the foundation of a separate manuscript in preparation (from the lab of first author C.S.K.) devoted to these glial cells.

Reviewer #1 (Recommendations for the authors):The authors should consider revising the title to give it encompass the overall conclusion and the context.

We felt it was important to emphasize in the title both the main approach we employed (scRNA-seq) and our major conclusions (extraordinary diversity of pulmonary neuroendocrine cells, signals, and potential targets in and outside the lung), and the title we settled on was constrained by journal limitations to character count.

In the Abstract section:It would be better to write the expanded form of the abbreviation at the first place throughout the text, for example, CGRP and GABA in the second line of the abstract.

Expanded forms of abbreviations have been added.

The authors' statement "release presumably regulated by the distinct, multimodal combinations of sensors expressed" needs to have specificity with examples. What sort of sensors?

This refers to the diverse sensors we found PNECs express (pp. 12-15), such as the well-studied mechanosensory ion channel (Piezo2). Figure 5 shows the full sensory gene and peptide expression profiles of all mouse PNECs (Figure 5A) and representative schematics of individual PNECs (Figure 5B). The word count limit of the Abstract prevents us from adding specific examples there.

The authors' statement "Expression of cognate receptors" should include specific examples.

Many examples are described in the text (pp. 9-12) and shown in Figure 4. The word count limit of the Abstract prevents us from adding specific examples there, however we changed "cognate receptors" to "peptide receptors" because use of the word "cognate" there could have been confusing.

In the Introduction section:Line 5: related functions should include a few specific examples.

We deleted "related functions" to keep this introductory statement simple.

Line 8-9: "..several neuropeptides including CGRP, which can signal locally to lung goblet (GABA) and immune cells (CGRP).." the abbreviation CGRP should be expanded at its first appearance. Also, the sentence should be re-phrased to explain the context of GABA and CGRP shown in brackets.

We added the expanded forms of the abbreviations GABA (γ-aminobutyric acid) and CGRP (calcitonin gene-related peptide) at their first appearance in the text. We added the word "via" in each set of brackets to clarify we mean here that PNECs signal locally in the lung to goblet cells via GABA, and to immune cells via CGRP.

Line 12: In the sentence, "they can be transformed by loss of tumor suppressor genes", please mention a few examples of the tumor suppressor genes.

The prominent tumor suppressors for small cell lung cancer are RB1 and TP53, as described in the cited references, but we chose not to single them out in the Introduction because our point here is only that neuroendocrine stem cells are a source of various tumors (in addition to their normal function as stem cells).

In the Results section:Section 2: The sentence "PNECs express dozens of neuropeptide and peptide hormone genes" does not seem to be scientifically sound. Authors should consider revising it to infer a concise sub-heading for this part.

Although we agree that the term "dozens" is inexact, we use it here because our scRNA-seq dataset of mouse PNECs identified thirty-one expressed peptidergic genes in mouse PNECs, and in a second dataset we identified 30 of those plus an additional 12. Hence the combined total for the two datasets shows 43 (47%) of the 91 peptidergic genes in mice are expressed in PNECs (Figure 1E, Supplementary file 4). On p. 7, we explain that 43 is likely a lower limit. We also identify 40 expressed peptidergic genes in our human PNEC scRNA-seq dataset (p. 17), and the details and expression values for each of the 91 neuropeptide genes analyzed across the 3 scRNA-seq datasets analyzed in mouse and human PNECs are provided in Supplementary file 4. We feel the term ‘dozens’ is appropriate here because it succinctly conveys this value in the sub-heading, without being overly precise about a value that depends on the sensitivity of the technique at detecting expression and how complete the PNEC dataset is.

Section 2: In the sentence "suggesting they can signal through serotonin imported from other sources.", the other source should be defined with precise citation, or is it just the authors' assumption?

Our suggestion of serotonin import by PNECs is simply an inference based on the detected expression of the serotonin reuptake transporter gene *Slc6a4* (but not serotonin biosynthetic genes *Tph1* and *Tph*2), as described in the first part of the sentence. The source(s) are not known but potentially include circulating serotonin, and we added this point about circulating serotonin (along with a relevant reference) on p. 5.

Section 2: Along with the names of genes, authors should consider citing the references vouching for neurotransmitters' type-specific genes in the sentence "PNECs may also use other neurotransmitters because rare cells expressed key dopaminergic (Th, Ddc), glutamatergic (Slc17a6), cholinergic (Slc18a3/VAChT), or histaminergic (Hdc) genes (Table S3).".

These are classic neurotransmitter biosynthesis and metabolism genes that are commonly used by neuroscientists to determine if a neuron expresses a particular neurotransmitter. We added a reference (Shammas, Hung, and Wang 2008) to the relevant tables (Supplementary files 3 and 8) to make this clear to the reader.

(A general comment/suggestion due to redundant appearance) The appearance of the phrase "dozens of genes" should be avoided throughout the text. Alternatively, authors can consider being specific about the numbers in their dataset.

We provide precise values in many places in the text (p. 6-7), figures (Figure 1E,F,G), and tables (Supplementary file 4), but as described above we feel the term ‘dozens’ succinctly conveys the value without being overly precise about a value that depends on the sensitivity of the technique at detecting expression and how complete the PNEC dataset is.

Authors are recommended to cite specific references for 17 PNEC peptidergic genes; based on what these genes have been selected?

Those 17 peptidergic genes are singled out because, as noted in the text (p. 6), they are the peptidergic genes that to our knowledge have not previously been detected in any type of neuroendocrine cell; we therefore cannot provide references for this type of negative data. However, the names and abbreviations of these and all other peptidergic genes in mice and human are provided in Supplementary file 5, along with the peptides they produce, the genes that encode receptor(s) for the peptides, and their major sites of expression and function of each peptide. We also provide summaries of our data describing their expression (Figure 1E, Supplementary file 4).

In the sentence "over an order of magnitude more than previously known", authors are recommended to include the previously known percentage or the number of peptidergic genes precisely.

CGRP and CHGA are the only previously established peptidergic genes in mouse PNECs, as noted on p. 6 and highlighted in Figure 1e (asterisk by gene names).

The Results sections 2 and 3 are overlapping. Authors should consider distinguishing the context.The authors should consider either combining them or segregating the talk about the peptidergic genes from the part of neuropeptide and peptide hormone genes.

Section 2 focuses on the total number and identity of peptidergic genes expressed in mouse PNECs, and Section 3 focuses on the distinct combinations of peptidergic genes expressed by individual mouse PNECs, highlighting PNEC cell diversity. Although these two sections could be merged into one large section, we thought it was instructive to separate the molecular (Section 2) and cellular (Section 3) conclusions.

In section 4, the sentence "PNECs are diverse" should have specificity about the type. Depending on what, the PNECs are diverse. To me, the results seem to be direct towards the PNEC diversity in the context of their functions. Authors should consider editing the sub-heading to depict a more specific context.

The word "diverse" here refers to sensory function, as indicated by the subheading title "PNECs are diverse, mutimodal sensors."

In section 5, again there are too many overlaps and redundancy of the information with sections 2 and 3. Authors should consider to re-arrange the result parts to give a concise and meaningful context.

Section 5 describes the corresponding properties of human PNECs. In writing the original manuscript, we tried merging the descriptions of the mouse and human data, but the comparison and contrast proved cumbersome. Hence, we ultimately chose to describe the human PNEC data in its own section (section 5), and we highlighted there both the conserved features as well as the distinct ones from mouse PNECs.

In section 6, the sentence "The human lung cell expression patterns of receptors for the 32 PNEC peptidergic signals with known receptors are shown in Figure S10A, identifying potential direct targets in lung." What are the direct targets the authors are referring to?

These refer to the predicted direct cellular targets of the peptidergic signal (as shown in Figure 6—figure supplement 3A), i.e. the lung cell type(s) that express the corresponding receptor for the peptide signal.

In section 6, the sentence "The predicted targets of the conserved PNEC signals were also largely conserved, for example, broad stromal and vascular targeting by inhibin, immune cell targeting by CGRP and VGF, and pericyte targeting by angiotensin.", the claim should also be supported with references.

These are new conclusions derived from the data and the analysis we present here, so there are no prior references to cite.

In section 7, the sentence "However, they did not recapitulate the full spectrum of PNEC diversity in their sensory and signaling gene profiles." Authors should consider explaining it briefly with specific citations.

This is shown in Figure 6D (compare pink bars to black dots), and we have added a call out to that panel right after that sentence to clarify this point. The following sentences provide specific examples.

Reviewer #2 (Recommendations for the authors):The manuscript would be considerably strengthened by the following:1. A rigorous definition of neuroendocrine cells should be stated clearly and applied consistently across all datasets. If cells are likely to be neuroendocrine but do not meet the complete definition (e.g., ZsGreen+ EpCAM- cells), then it might be helpful to indicate them in the figure(s) and explain how they do or do not differ in their properties from the rigorously-defined subset of PNECs.2. The presentation of sensitivity and specificity of PNEC gene expression would benefit from a different graphical format. For example, some type of two-dimensional plots might illustrate these points about peptidergic gene expression more clearly.3. Greater clarity on the number of samples examined and the addition of statistical analyses would be very helpful to include.

We thank the reviewer for their careful reading and suggestions for the manuscript. The suggestions have been incorporated.

Reviewer #3 (Recommendations for the authors):This study catalogs the transcriptomic diversity of PNECs. Agt action on PSN neurons was included as a functional test. Additional functional experiments will enhance the impact of the study and warrant publication in this journal.1. Pointing out which genes are expressed in PNECs but not in TNECs will be informative.

In the original manuscript we noted tracheal neuroendocrine cells (TNEC) markers (from the mouse tracheal cell transcriptomic atlas by Montoro et al., Nature 2018) that were not expressed in our profiled mouse PNECs. The converse (PNEC markers not detected in TNECs) are also of interest but are more problematic to identify from the available data because the droplet-based scRNA-seq method used for profiling TNECs (in Montoto et al) is subject to greater technical dropout than the plate-based profiling method we used for PNECs. We analyzed the expression profiles for mouse tracheal cells isolated from wild-type mice Montoro et al. (‘GSE103354_Trachea_dropblet_UMIcounts.txt’ in GEO), selecting only the TNECs that clustered distinctly from the other tracheal epithelial cells (69 TNECs) and eliminating transitional cells. We then searched for expression in these TNECs of the most robustly-expressed PNEC marker genes in our dataset, and found one gene (*Igf2*) with no expression in TNECs and others (*Piezo2, Ptprn, Meg3*) with very low or sparse expression. We added these apparent PNEC-selective markers to the text (p. 5) and to Figure 5—figure supplement 1F and Supplementary file 1, however this designation remains provisional (due to the technical dropout rate of the method used for profiling TNECs) until it is confirmed by direct comparison of their expression in PNECs vs. TNECs.

2. The statement that most peptidergic genes are "expressed in rare PNEC subpopulations" needs to be based on smFISH data, and not just scRNA-seq data given the drop-off effects. The lack of reads in the expected potassium channels also warrants further investigation by smFISH.

See response to this comment above (Essential Revision point 5).

3. Neuropeptides are typically enriched in vesicles on the basal side of PNECs, and this is reflected in most of the staining results. However, in Figure 3C, this does not appear to be the case, which warrants re-investigation.

See response to this comment above (Essential Revision point 6).

4. In Figure 4F, neuronal traces in the lower panel have 3 different colors (grey, dark grey, and black), what are the numbers of cells in each group? And what may explain why Agtr1a+ PSN7 neurons were not activated after Agt II application? Are these neurons activated at higher concentrations?

See response to this comment above (Essential Revision point 7).

5. Agt is expressed by other cell types in the lung. Do Agtr1a+ neurons only target PNECs? What is the evidence that PNECs are the source of Agt that activates Agtra1 neurons?

See response to this comment above (Essential Revision point 8).

6. Are there expression pattern differences between solitary and clustered PNECs in either mouse or human?

See response to this comment above (Essential Revision point 9).

7. Is there any information in the single cell data that would explain the cause for rare PNEC types being amplified in the carcinoid?

See response to this comment above (Essential Revision point 10).

8. Description of signature and staining localization of glial cells will be informative.

See response to this comment above (Essential Revision point 11).